# CREATIVE SKETCH GENERATION

**Songwei Ge** [*]
University of Maryland, College Park
songweig@umd.edu

**Vedanuj Goswami & C. Lawrence Zitnick**
Facebook AI Research
{vedanuj,zitnick}@fb.com

**Devi Parikh**
Facebook AI Research
Georgia Institute of Technology
parikh@gatech.edu

## ABSTRACT

Sketching or doodling is a popular creative activity that people engage in. However, most existing work in automatic sketch understanding or generation has focused on sketches that are quite mundane. In this work, we introduce two datasets of creative sketches – Creative Birds and Creative Creatures – containing 10k sketches each along with part annotations. We propose DoodlerGAN – a part-based Generative Adversarial Network (GAN) – to generate unseen compositions of novel part appearances. Quantitative evaluations as well as human studies demonstrate that sketches generated by our approach are more creative and of higher quality than existing approaches. In fact, in Creative Birds, subjects prefer sketches generated by DoodlerGAN over those drawn by humans!

## 1 INTRODUCTION

*The true sign of intelligence is not knowledge but imagination.* – *Albert Einstein*

From serving as a communication tool since prehistoric times to its growing prevalence with ubiquitous touch-screen devices – sketches are an indispensable visual modality. Sketching is often used during brainstorming to help the creative process, and is a popular creative activity in itself.

Sketch-related AI so far has primarily focused on mimicking the human ability to perceive rich visual information from simple line drawings (Yu et al., 2015; Li et al., 2018) and to generate minimal depictions that capture the salient aspects of our visual world (Ha & Eck, 2018; Isola et al., 2017). Most existing datasets contain sketches drawn by humans to realistically mimic common objects (Eitz et al., 2012; Sangkloy et al., 2016; Jongejan et al., 2016; Wang et al., 2019).

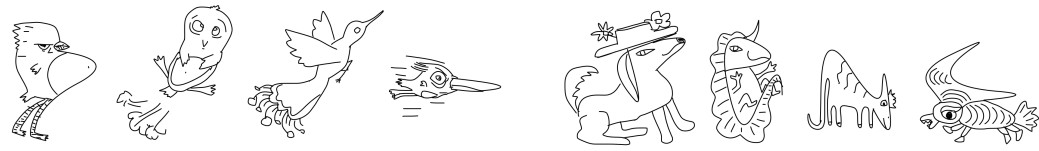

Figure 1: Cherry-picked example sketches from our proposed datasets: Creative Birds (left) and Creative Creatures (right). See random examples in Figure 2 and Figures 12 and 13 in the Appendix.

In this work we focus on *creative* sketches. AI systems that can generate and interpret creative sketches can inspire, enhance or augment the human creative process or final artifact. Concrete scenarios include automatically generating an initial sketch that a user can build on, proposing the next set of strokes or completions based on partial sketches drawn by a user, presenting the user with possible interpretations of the sketch that may inspire further ideas, etc.

---

[*]The work was done when the first author interned at Facebook AI Research.

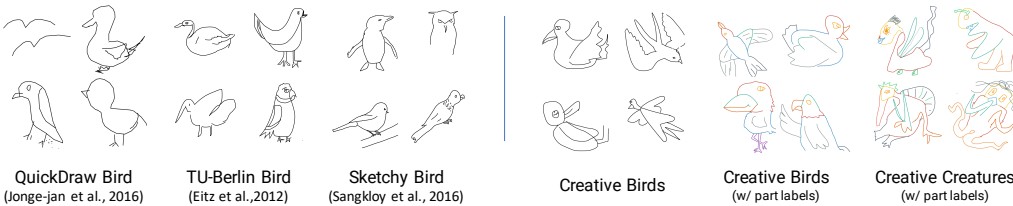

QuickDraw Bird
(Jonge-jan et al., 2016)

TU-Berlin Bird
(Eitz et al.,2012)

Sketchy Bird
(Sangkloy et al., 2016)

Creative Birds

Creative Birds
(w/ part labels)

Creative Creatures
(w/ part labels)

Figure 2: Random sketches from existing datasets (left) and our creative sketches datasets (right).

AI for creative sketches is challenging. They are diverse and complex. They are unusual depictions of visual concepts while simultaneously being recognizable. They have subjective interpretations like aesthetics and style, and are semantically rich – often conveying a story or emotions.

To facilitate progress in AI-assisted creative sketching, we collect two datasets – Creative Birds and Creative Creatures (Figure 1) – containing 10k creative sketches of birds and generic creatures respectively, along with part annotations (Figure 2 right columns). To engage subjects in a creative exercise during data collection, we take inspiration from a process doodling artists often follow. We setup a sketching interface where subjects are asked to draw an eye arbitrarily around a random initial stroke generated by the interface. Subjects are then asked to imagine a bird or generic creature that incorporates the eye and initial stroke, and draw it one part at a time. Figure 2 shows example sketches from our datasets. Notice the larger diversity and creativity of birds in our dataset than those from existing datasets with more canonical and mundane birds.

We focus on creative sketch generation. Generating novel artifacts is key to creativity. To this end we propose DoodlerGAN – a part-based Generative Adversarial Network (GAN) that generates novel part appearances and composes them in previously unseen configurations. During inference, the model automatically determines the appropriate order of parts to generate. This makes the model well suited for human-in-the-loop interactive interfaces where it can make suggestions based on user drawn partial sketches. Quantitative evaluation and human studies show that our approach generates more creative and higher quality sketches than existing approaches. In fact, subjects prefer sketches generated by DoodlerGAN over human sketches from the Creative Birds dataset!

Our datasets, code, and a web demo are publicly available [1].

## 2 RELATED WORK

Sketches have been studied extensively as a visual modality that is expressive yet minimal. The sparsity of sketches compared to natural images has inspired novel modelling techniques. We discuss existing sketch datasets and sketch generation approaches. Other related work includes sketch recognition (Yu et al., 2015; Li et al., 2018), sketch-based image retrieval (Yu et al., 2016; Liu et al., 2017; Ribeiro et al., 2020) and generation (Gao et al., 2020; Lu et al., 2018; Park et al., 2019). An overview of deep learning approaches for sketches can be found in this survey (Xu et al., 2020)

**Sketch datasets.** Existing sketch datasets such as TU-Berlin (Eitz et al., 2012), Sketchy (Sangkloy et al., 2016), ImageNet-Sketch (Wang et al., 2019) and QuickDraw (Jongejan et al., 2016) are typically focused on realistic and canonical depictions of everyday objects. For instance, sketches in the Sketchy dataset (Sangkloy et al., 2016) were drawn by humans mimicking a natural image. Sketches in the QuickDraw dataset (Jongejan et al., 2016) were collected in a Pictionary-like game setting – they were drawn under 20 seconds to be easily recognized as a target object category. This is in stark contrast with how people engage in doodling as a creative activity, where they take their time, engage their imagination, and draw previously unseen depictions of visual concepts. These depictions may be quite unrealistic – including exaggerations or combinations of multiple categories. Our datasets contain such creative sketches of birds and generic creatures. See Figures 1 and 2. Our data collection protocol was explicitly designed to engage users in a creative process. Also note that while not the focus of this paper, our datasets are a valuable resource for sketch segmentation approaches. See Section A in the Appendix for further discussion.

---

[1] songweige.github.io/projects/creative_sketech_generation/home.html

**Sketch generation.** Earlier studies generated a sketch from an image via an image-to-image translation approach (Isola et al., 2017; Song et al., 2018; Li et al., 2019b). Subsequently, fueled by the release of large-scale free-hand sketch datasets, methods that generate human-like sketches from scratch have interested many researchers. One of the early works in this direction was SketchRNN (Ha & Eck, 2018) – a sequence-to-sequence Variational Autoencoder (VAE) that models the temporal dependence among strokes in a sketch. Later approaches (Chen et al., 2017; Cao et al., 2019) incorporated a convolutional encoder to capture the spatial layout of sketches. Reinforcement learning has also been studied to model the sequential decision making in sketch generation (Zhou et al., 2018; Huang et al., 2019; Mellor et al., 2019). In contrast with generating entire sketches, sketch completion approaches generate missing parts given a partial sketch. Sketch-GAN (Liu et al., 2019) adopted a conditional GAN model as the backbone to generate the missing part. Inspired by the large-scale language pretraining approaches (Devlin et al., 2019), Sketch-BERT (Lin et al., 2020) learns representations that capture the "sketch gestalt". We propose a part-based approach, DoodlerGAN, for sketch generation. Compared to previous approaches (Ha & Eck, 2018; Cao et al., 2019) that learn to mimic how humans draw, our goal is to create sketches that were not seen in human drawings. The idea of generating different components as parts of the overall sketch can be traced back to compositional models (Chen et al., 2006; Xu et al., 2008). We explore these ideas in the context of deep generative models for sketches. Also relevant are approaches that exploit spatial structure when generating natural images – foreground vs. background (Yang et al., 2017) or objects and their spatial relationships (Johnson et al., 2018). A discussion on creative tools based on sketch generation can be found in Section B the Appendix.

## 3 CREATIVE SKETCHES DATASETS

To facilitate progress at the intersection of machine learning and artificial creativity in the context of sketches, we collected two datasets: Creative Birds and Creative Creatures. The first is focused on birds, and the second is more diverse and challenging containing a variety of creatures.

### 3.1 DATASET COLLECTION

Both datasets were collected on Amazon Mechanical Turk using a sketching web interface[2]. In order to engage subjects in a creative exercise, and encourage diversity across sketches, the interface generates a random initial stroke formed by connecting $K$ keypoints on the canvas via Bezier curves. These points were picked via heuristics such that the strokes are smooth with similar lengths and have limited self occlusion (see Figure 9 in the Appendix). Next, inspired by a process doodling artists (e.g., Michal Levy) use, subjects were told to add an eye to the canvas wherever they like. They were then asked to step back, and visualize how the initial stroke and the eye can be incorporated in a creative sketch of a bird or arbitrary creature (depending on the dataset).

Subjects were then asked to draw one part of the bird or creature at a time, indicating via a drop down menu which part they were drawing. Each part can be drawn using multiple strokes. For Creative Birds, the options include 7 common parts of birds (head, body, beak, tail, mouth, legs, wings). For Creative Creatures, the 16 parts (e.g., paws, horn, fins, wings) cover terrestrial, aquatic, and aerial creatures. In our pilot studies we found that providing subjects this structure of drawing one part at a time increased the quality of sketches, and giving subjects the option to pick parts themselves made the exercise more natural and increased the diversity of sketches. As a by product, each stroke has a corresponding part annotation (see Figure 2), which can be a rich resource for sketch segmentation and part-based sketch recognition or generation models. Subjects are required to add at least five parts to the sketch (in addition to the eye and initial stroke) before they can submit the sketch.

Once done with adding parts, subjects were given the option to add additional details to the sketch. Finally, they were asked to add a free-form natural language phrase to title their sketch to make it more expressive. In this work we do not use the details and phrases, but they may be useful in the future for more detailed sketch generation and automatic creative sketch interpretation. Except for Figure 1, all example sketches shown in the paper are without the details unless noted otherwise.

In addition to contributing to the creativity and diversity of sketches, the initial random strokes also add constraints, making sketch creation more challenging. Subjects have more constraints

---

[2]Our interface can be seen at `https://streamable.com/jt4sw1`

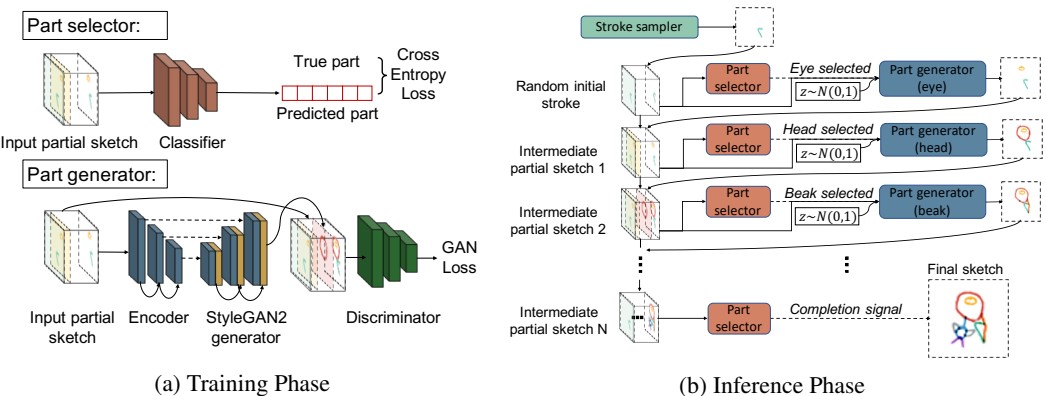

(a) Training Phase           (b) Inference Phase

Figure 3: DoodlerGAN. (a) During training, given an input partial sketch represented as stacked part channels, a part selector is trained to predict the next part to be generated, and a part generator is trained for each part to generate that part. (b) During inference, starting from a random initial stroke, the part selector and part generators work iteratively to complete the sketch.

when asked to draw only birds in Creative Birds, so we give them less starting constraints ($K = 3$ keypoints in the initial strokes), but for Creative Creatures subjects have more freedom in the animals they draw so we provide more constraints ($K = 6$). Example initial strokes for both datasets and some insights from pilot data collection can be found in Figure 9 and Section C in the Appendix.

## 3.2 DATA ANALYSIS

We collected 10k sketches for both datasets. Filtering out sketches from workers who did not follow the instructions well, we have 8067 and 9097 sketches in Creative Birds and Creative Creatures respectively. See Figure 2 for random examples of sketches from both datasets. For comparison, we also show birds from other existing datasets. Notice that the sketches in our datasets are more creative, and sketches from Creative Creatures are more diverse and complex than those in Creative Birds. We conducted a human study comparing 100 sketches from Creative Birds to 100 from QuickDraw across 5 subjects each. Our sketches were rated as more creative 67% of the time. More example sketches from our datasets can be found in Figures 12 and 13 in the Appendix. In Figure 14 in the Appendix we also show examples where different sketches incorporate similar initial strokes, suggesting room for creativity and variation that a generative model could learn. Sketches of individual parts are shown in Figure 10 and 11 in the Appendix. Analysis of the part annotations, including the order in which parts tend to be drawn, is provided in Section D in the Appendix.

## 4 DOODLERGAN: A PART-BASED SKETCH GENERATION MODEL

Objects are typically a configuration of parts (e.g., animals have two or four legs, a mouth below two eyes). Moreover, humans often sketch by drawing one part at a time. We approach creative sketch generation by generating novel appearances of parts and composing them in previously unseen configurations. Another benefit of parts is that creative sketches exhibit large diversity in appearance, but this complexity is significantly reduced if the sketches are decomposed into individual parts.

Our approach, DoodlerGAN, is a part-based GAN that sequentially generates one part at a time, while ensuring at each step that the appearance of the parts and partial sketches comes from corresponding distributions observed in human sketches. Humans do not draw parts in the same order across sketches, but patterns exist. To mimic this, and to adapt well to a (future) human-in-the-loop setting, DoodlerGAN automatically determines the order of parts. Concretely, DoodlerGAN contains two modules: the part generator and the part selector, as shown in Figure 3. Given a part-based representation of a partial sketch, the part selector predicts which part category to draw next. Given a part-based representation of a partial sketch and a part category, the part generator generates a raster image of the part (which represents both the appearance and location of the part).

## 4.1 ARCHITECTURE

We represent sketches as raster images (i.e., grids of pixels) as opposed to vector representations of strokes. This makes detailed spatial information readily available, which helps the model better assess where a part connects with the rest of the sketch. A vector representation captures the low-level sequence, which may not be relevant to the overall quality of the generated sketch (e.g., a circular head will look the same whether it is drawn clockwise or counter-clockwise). Our part-based approach models the sequence at an appropriate level of abstraction (parts).

The part generator is a conditional GAN based on the StyleGAN2 architecture (Karras et al., 2019; 2020). We generate sketches at a $64 \times 64$ resolution. We use a 5-layer StyleGAN2 generator with $[512, 256, 128, 64, 32]$ output channels and starting from a $4 \times 4 \times 64$ constant feature map. To encode the input partial sketch, we use a 5-layer CNN with $[16, 32, 64, 128, 256]$ output channels. Each layer contains two convolutional layers with $3 \times 3$ kernels followed by a LeakyRelu activation with a negative slope of $0.2$. Inspired by the design of U-Net (Ronneberger et al., 2015), we downsample the intermediate feature map after each encoder layer and concatenate it channel-wise with the corresponding layers in the generator. See dashed lines in Figure 3a. This gives the generator access to the hierarchical spatial structure in the input sketch.

The input partial sketch is represented by stacking each part as a channel, along with an additional channel for the entire partial sketch. If a part has not been drawn, the corresponding channel is a zero image. We find a part-based representation to be crucial for predicting appropriate part locations. For instance, without the part channels, the generated legs were often not connected to the body.

Following StyleGAN2, we borrow the discriminator architecture from (Karras et al., 2018). We give the discriminator access to the input partial sketch and the corresponding part channels, so that the discriminator is able to distinguish fake sketches from real ones not just based on the appearance of the part, but also based on whether the generated part is placed at an appropriate location relative to other parts (e.g., heads are often around the eyes). Specifically, similar to the generator, the input to the discriminator is an image with (number of parts + 1) channels. In Figure 3a, the difference between the real and fake data fed into the discriminator occurs in the red channel where the fake data contains the generated part (as opposed to the real one), and the last white channel where the generated (vs. real) part is combined with the input partial sketch using a max operation.

The generator and discriminator parameters are not shared across parts. In our experiments, a unified model across parts failed to capture the details in parts such as wings. We also experimented with fine tuning the model end-to-end but generally observed inferior generation quality.

For the part selector, we use the same architecture as the encoder but with a linear layer added at the end to produce logits for different parts. The part selector is also expected to decide when the sketch is complete and no further parts need to be generated. Therefore, the output dimension of the linear layer is set to (number of parts + 1). We convert the human sketches in our dataset to (partial sketch, next-part) pairs and use that to train the part selector in a supervised way.

## 4.2 TRAINING AND INFERENCE

With strong conditional information, the discriminator in the part generator gets easily optimized and consequently provides zero gradient for the generator. As a result, the generator gets stuck in the early stages of training and only generates the zero image afterwards. We introduce a sparsity loss as an additional regularizer, which is the $L2$ norm between the sum of pixels in the generated and real parts, encouraging the model to generate parts of similar sizes as the real parts. This helps stabilize the training and helps the generator learn even when little signal is provided by the discriminator. During training, we also augment the training sketches by applying small affine transformations to the vector sketch images before converting them to raster images. During inference, we follow the same procedure as data collection: we first automatically sample an initial stroke from our interface (see Figure 9) and then use the part selector and part generators iteratively to complete the sketch.

Conditional GANs have been shown to not be sensitive to noise, generating the same sample for different noise vectors (Zhu et al., 2017; Donahue et al., 2017). We encountered similar issues, especially for parts drawn at a later stage when more conditional information is available. This is expected; the more complete the partial sketch, the fewer ways there are to reasonably complete it. We found that our model is sensitive to minor perturbations to the input partial sketch. To increase

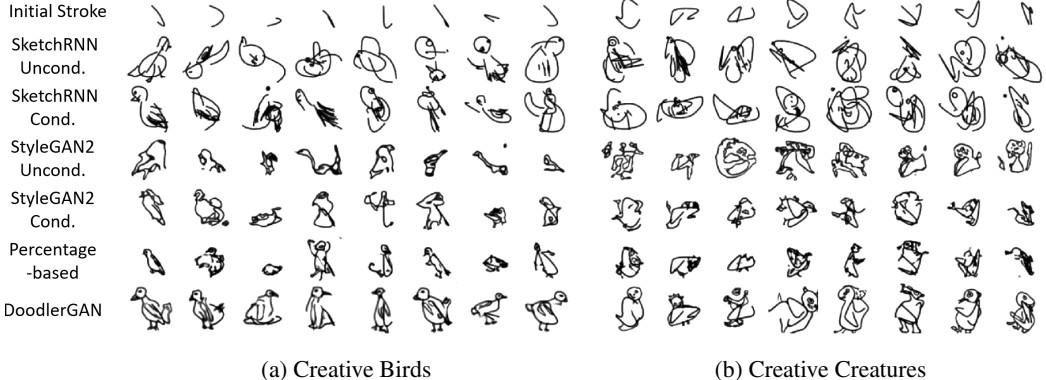

|  |  |
|---|---|
| (a) Creative Birds | (b) Creative Creatures |

Figure 4: Random generations from each approach. DoodlerGAN generations are noticeably higher quality compared to baselines. More DoodlerGAN generations are in Figure 15 in the Appendix.

the diversity in generations, we propose a simple trick we call **conditioning perturbation**. We apply random translations sampled from $\mathcal{N}(0, 2)$ to the input partial sketch to generate multiple parts and then translate the generated parts back to align with the original partial sketch.

For further details of training, data augmentation and inference see Section F in the Appendix.

## 5 RESULTS

We quantitatively and qualitatively (via human studies) evaluate our approach compared to strong baselines and existing state-of-the-art approaches trained on our creative datasets:

**StyleGAN2 Unconditional.** We train StyleGAN2 using the same hyperparameters and data augmentation settings used for DoodlerGAN, to generate the entire sketches in one step. To avoid mode collapse, we add the minibatch discriminator layer (Salimans et al., 2016) in its discriminator. This represents a state-of-the-art approach in image generation.

**StyleGAN2 Conditional.** This approach is the same as the one described above, but we condition the one-step generation on the initial stroke (encoded using the same encoder as in DoodlerGAN).

**SketchRNN Unconditional.** We use the SketchRNN model (Ha & Eck, 2018) trained on our datasets. This represents the state-of-the-art in sketch generation. We optimized the architecture (encoder, decoder and latent space sizes, temperature $\gamma$) and used heuristic post processing to eliminate obvious failure cases to adapt this approach the best we could to our datasets.

**SketchRNN Conditional.** This approach is SketchRNN described above, except during inference we fix the first few points and the pen states based on the random initial stroke. These are fed as input to continue the sequential generation of the sketch.

**Percentage-based.** This approach is DoodlerGAN, except instead of using parts, we divide the sketch into 20% chunks (based on the order in which strokes were drawn) and use them as "parts". A comparison to this approach allows us to demonstrate the effectiveness of semantic parts.

Randomly selected generations from each approach are shown in Figure 4. In StyleGAN2 generations we can see a general bird contour but clean details are lacking. In SketchRNN we can see the generated eye but later strokes do not form a coherent sketch. The failure to generate complex sketches with SketchRNN has also been reported in (Ha & Eck, 2018). DoodlerGAN generates noticeably higher quality sketches, with different parts of the sketch clearly discernible. More sketches generated by DoodlerGAN can be found in Figure 15.

In human evaluation, we also compare bird generations from DoodlerGAN trained on Creative Birds to the *human-drawn* bird sketches from QuickDraw to represent the "best" (oracle) generator one can hope to get by training on QuickDraw. (A natural equivalent for Creative Creatures is unclear.) In addition to the five generation-based methods, we also compare our method with a strong retrieval-based method in Section J in the Appendix.

To evaluate, we generate $10,000$ sketches using all approaches based on a randomly sampled set of previously unseen initial strokes (for the conditional approaches).

## 5.1 QUANTITATIVE EVALUATION

For quantitative evaluation, we consider two metrics used in previous studies: Fréchet inception distances (FID) (Heusel et al., 2017) and generation diversity (GD) (Cao et al., 2019). For this, we trained an Inception model on the QuickDraw3.8M dataset (Xu et al., 2020). See Section G in the Appendix for more training details. In Table 1 we see that DoodlerGAN has the best FID score relative to other approaches, while maintaining similar diversity.

Table 1: Quantitative evaluation of DoodlerGAN against baselines on Fréchet inception distances (FID), generation diversity (GD), characteristic score (CS) and semantic diversity score (SDS). The methods with the best scores are in **bold**.

| Methods | Creative Birds | | | Creative Creatures | | | |
|---|---|---|---|---|---|---|---|
| | FID($\downarrow$) | GD($\uparrow$) | CS($\uparrow$) | FID($\downarrow$) | GD($\uparrow$) | CS($\uparrow$) | SDS($\uparrow$) |
| Training Data | - | 19.40 | 0.45 | - | 18.06 | 0.60 | 1.91 |
| SketchRNN Uncond. | 79.57 | 17.19 | 0.20 | 60.56 | 15.85 | 0.43 | 1.22 |
| SketchRNN Cond. | 82.17 | 17.29 | 0.18 | 54.12 | **16.11** | 0.48 | 1.34 |
| StyleGAN2 Uncond. | 71.05 | **17.49** | 0.23 | 103.24 | 14.41 | 0.18 | 0.72 |
| StyleGAN2 Cond. | 130.93 | 14.45 | 0.12 | 56.81 | 13.96 | 0.37 | 1.17 |
| Percentage-based | 103.79 | 15.11 | 0.20 | 57.13 | 13.86 | 0.41 | 1.17 |
| DoodlerGAN | **39.95** | 16.33 | **0.69** | **43.94** | 14.57 | **0.55** | **1.45** |

We also introduce two additional metrics. The first is the characteristic score (CS) that checks how often a generated sketch is classified to be a bird (for Creative Birds) or creature (for Creative Creatures) by the trained Inception model. The higher the score – the more recognizable a sketch is as a bird or creature – the better the sketch quality. The second is the semantic diversity score (SDS) that captures how diverse the sketches are in terms of the different creature categories they represent (this is more meaningful for the Creative Creatures dataset). Higher is better. Note that GD captures a more fine-grained notion of diversity. For instance, if all generated sketches are different dog sketches, GD would still be high but SDS would be low. See Section H in the Appendix for further details about these two metrics. In Table 1 we see that DoodlerGAN outperforms existing approaches on both metrics. In fact, DoodlerGAN even outperforms the human sketches on the Creative Birds dataset! This trend repeats in human evaluation.

## 5.2 HUMAN EVALUATION

Automatic evaluation of image generation and creative artifacts are open research questions. Therefore, we ran human studies on Amazon Mechanical Turk (AMT). Specifically, we showed subjects pairs of sketches – one generated by DoodlerGAN and the other by a competing approach – and ask which one (1) is more creative? (2) looks more like a bird / creature? (3) they like better? (4) is more likely to be drawn by a human? For the conditional baselines, we also ask (5) in which sketch is the initial stroke (displayed in a different color) better integrated with the rest of the sketch? We evaluated 200 random sketches from each approach. Each pair was annotated by 5 unique subjects.

Figure 5 shows the percentage of times DoodlerGAN is preferred over the competing approach. We also plot the Bernoulli confidence intervals ($p = 0.05$). Values outside the band are statistically significantly different than 50%, rejecting the null hypothesis that both approaches are equally good. For Creative Birds, DoodlerGAN significantly outperforms the five baselines on all five questions. It even beats the real human-drawn sketches, not just from QuickDraw, but also from our Creative Birds dataset on most dimensions! Creative Creatures is a more challenging dataset. DoodlerGAN outperforms other approaches but not human sketches. DoodlerGAN is not statistically significantly better (or worse) than SketchRNN Conditional in terms of creativity. Note that creativity in itself may not be sufficient – an arbitrary pattern of strokes may seem creative but may not be recognized as a bird or creature. The combination of creativity and looking like a bird or creature is a more

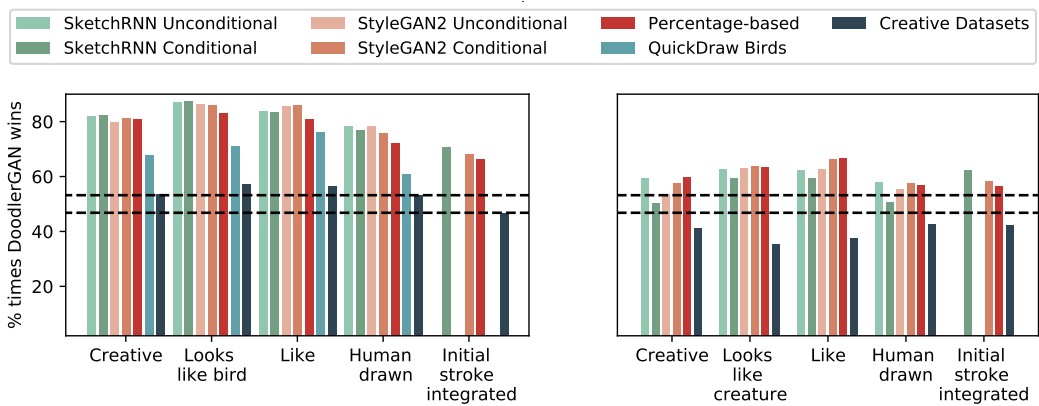

Figure 5: Human evaluation results on Creative Birds (left) and Creative Creatures (right). Higher values → DoodlerGAN is preferred over the approach more often.

holistic view, which we believe is better captured by the overall metric of which sketch subjects like better. DoodlerGAN statistically significantly outperforms SketchRNN on that metric. Similarly, subjects can not differentiate between DoodlerGAN and SketchRNN Conditional in terms of being more likely to be drawn by humans. Note that unlike some other AI tasks (e.g., recognizing image content), an average human need not be the gold standard for creative tasks such as sketching.

## 5.3 NOVELTY

Figure 16 in the Appendix shows generated sketches and their nearest neighbor training sketches using Chamfer distance. We see that the closest training sketches are different, demonstrating that DoodlerGAN is generating previously unseen creative sketches. In fact, by evaluating a sketch classification model trained on QuickDraw on our generated Creative Creatures sketches, we identify several sketches that have a high response ($>0.25$) to more than one category. We find that our model has generated hybrids of a penguin and mouse, a panda and a snake, a cow and a parrot (see Figure 6) – a strong indication of creativity (Boden, 1998; Yu & Nickerson, 2011)!

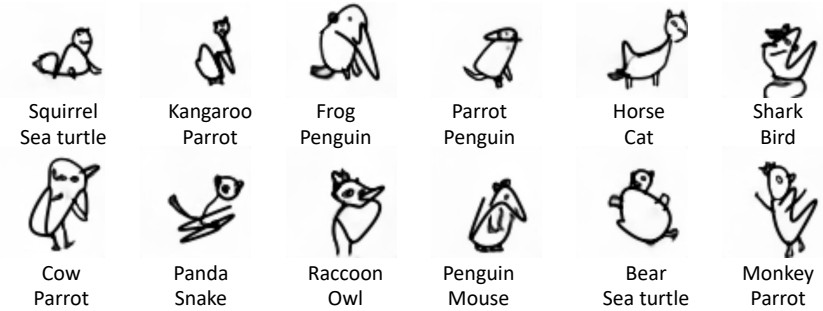

Figure 6: DoodlerGAN generates sketches of hybrid creatures

In the Appendix, we include several other evaluation studies to demonstrate (1) DoodlerGAN's ability to generate a diverse set of parts in an interpretable fashion where smoothly varying the input noise to one layer changes the part shape and to another layer changes the position (Section I) (2) DoodlerGAN's ability to complete a sketch better than a retrieval-based approach (Section J) (3) the improvements in sketch quality due to the different design choices made in the part generator and selector (Section K). Finally, we also briefly discuss a heuristic method to convert the generated raster images to a vector representation (Section L) that enables rendering sketches at arbitrary resolutions. This conversion process adds an unintended but delightful aesthetic to the sketches! In human studies we find that subjects prefer this aesthetic to the raw generated images 96% of the times. We also ran all our human studies with this aesthetic and find that similar trends hold.

## 6 CONCLUSION

In this work we draw attention to creative sketches. We collect two creative sketch datasets that we hope will encourage future work in creative sketch interpretation, segmentation, and generation. In this paper we focus on the latter, and propose DoodlerGAN – a part-based GAN model for creative sketch generation. We compare our approach to existing approaches via quantitative evaluation and human studies. We find that subjects prefer sketches generated by DoodlerGAN. In fact, for birds, subjects prefer DoodlerGAN's sketches over those drawn by humans! There is significant room for improvement in generating more complex creative sketches (e.g., Creative Creatures). Future work also involves exploring human-machine collaborative settings for sketching.

### ACKNOWLEDGMENTS

The Georgia Tech effort was supported in part by AFRL, DARPA, ONR YIP and Amazon. The views and conclusions contained herein are those of the authors and should not be interpreted as necessarily representing the official policies or endorsements, either expressed or implied, of the U.S. Government, or any sponsor.

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

## A    SKETCH SEGMENTATION DATASETS

Previous sketch segmentation datasets (Schneider & Tuytelaars, 2016; Yang et al., 2020; Li et al., 2019a; Wu et al., 2018) are too sparsely annotated to effectively train deep models. For example, the SPG dataset (Li et al., 2019a) contains 800 sketches per category with part annotations. Note that most approaches train one model per category. The SketchSeg (Wu et al., 2018) dataset contains 100 labeled sketches per category, then further augmented by automatically generating more sketches with a trained SketchRNN (Ha & Eck, 2018) model. Our proposed datasets are exhaustively annotated – all 10k sketches in both datasets have part annotations.

## B    EXISTING CREATIVE TOOLS BASED ON SKETCH GENERATION

The growing availability of sketch generation approaches has spurred a variety of creative applications and art projects. The book Dreaming of Electric Sheep (Diaz-Aviles, 2018) inspired by Philip K. Dick's famous book contains 10,000 sheep sketches generated by DCGAN (Radford et al., 2015) trained on the QuickDraw dataset (Jongejan et al., 2016). Quick, Draw! (quickdraw.withgoogle.com) recognizes what a user is drawing as they draw it. Auto-Draw (Lab, 2017) returns high quality illustrations or sketches created by experts that match a novice's sketch. Edges2cats (Hesse, 2017) is an online tool based on pix2pix (Isola et al., 2017) that converts free-hand drawings to realistic images. Our dataset and approach can be used to boost more applications related to draw a creative sketch.

## C    INSIGHTS FROM PILOT DATA COLLECTION PILOT

We found that when using a shorter initial stroke, the generic creatures were often recognizable as specific animals. But with longer strokes, we more frequently found fictional creatures that are high quality (look like creatures) but not recognizable as a real animal, indicating higher creativity. Piloting longer initial strokes with birds indicated that subjects had trouble incorporating the initial stroke well while maintaining coherence. Shorter strokes led to simpler sketches (still significantly more complex and creative than QuickDraw, Figure 2), making Creative Birds a more approachable first dataset for creative sketch generation, and Creative Creatures a notch above in difficulty.

## D    STROKE- AND PART-LEVEL ANALYSIS

We compare our Creative Birds and Creative Creatures datasets to previous datasets in Table 2. As a proxy for complexity, we report the average stroke length (normalized by the image size) and number of strokes across the bird sketches in each dataset. We see that our datasets have the longest and most number of strokes. The difference is especially stark w.r.t. QuickDraw (twice as long and three times as many strokes), one of the most commonly used sketch datasets.

Table 2: Stroke statistics

|  | normalized stroke length | number of strokes |
|---|---|---|
| Sketchy | $5.54 \pm 2.02$ | $15.57 \pm 10.54$ |
| Tu-Berlin | $5.46 \pm 1.75$ | $14.94 \pm 10.18$ |
| QuickDraw | $4.00 \pm 2.97$ | $6.85 \pm 4.19$ |
| Creative Birds (Ours) | $7.01 \pm 2.89$ | $20.74 \pm 13.37$ |
| Creative Creatures (Ours) | $8.40 \pm 3.47$ | $22.65 \pm 14.73$ |

Next we analyze the part annotations in our datasets. Table 3 shows the number of sketches containing each of the parts. Notice that each part in Creative Creatures has significantly fewer examples (in the extreme case, only 302 for paws.) than most parts in Creative Birds, making Creative Creatures additionally challenging. Most parts in Creative Birds are present in more than half the sketches except for the mouth, which we find is often subsumed in beak. For Creative Creatures, the numbers of sketches containing different parts has higher variance. Common parts like body and head have similar occurrence in both datasets. Example parts are shown in Figures 10 and 11. Note that the initial stroke can be used as any (entire or partial) part, including body but will not be annotated as such. This means that sketches without an annotated body likely still contain a body.

In Figures 7 and 8 we analyze the order in which parts tend to be drawn in both datasets.

## E    EXAMPLE SKETCHES, PARTS, AND INITIAL STROKES

See Figures 12 and 13 for example sketches from the Creative Birds and Creative Creatures datasets respectively. See Figures 10 and 11 for example parts from both datasets. See Figure 9 for examples of the initial strokes used in the data collection and sketch generation process.

Table 3: Part statistics

| Creative Birds | total | eye | wings | legs | head | body | tail | beak | mouth |
|---|---|---|---|---|---|---|---|---|---|
| | 8067 | 8067 | 7764 | 5564 | 7990 | 8477 | 5390 | 7743 | 1447 |

| Creative Creatures | total | eye | arms | beak | mouth | body | ears | feet | fin |
|---|---|---|---|---|---|---|---|---|---|
| | 9097 | 9097 | 2862 | 2163 | 6143 | 7490 | 3009 | 2431 | 1274 |
| | hands | head | hair | horns | legs | nose | paws | tail | wings |
| | 1771 | 6362 | 3029 | 1738 | 3873 | 3428 | 302 | 3295 | 1984 |

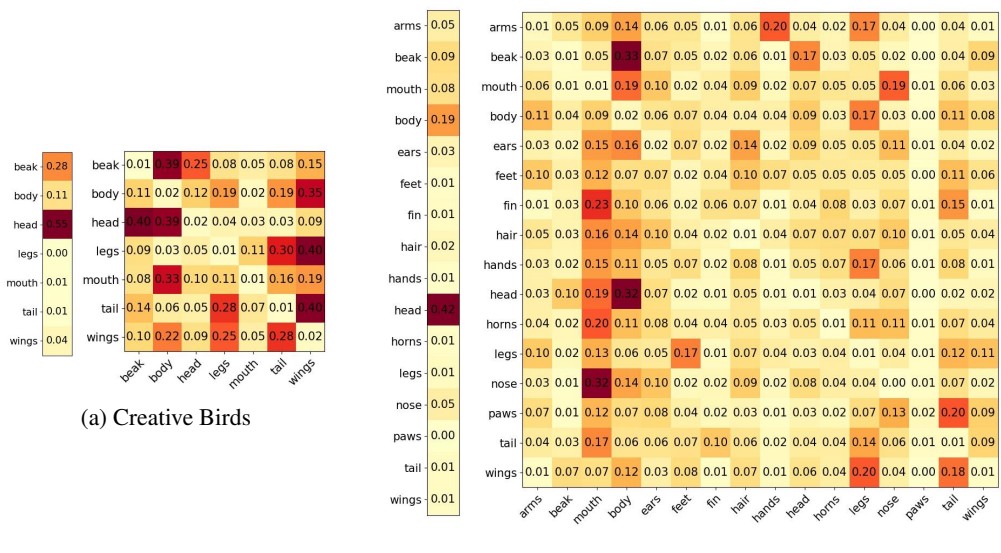

(a) Creative Birds

(b) Creative Creatures

Figure 7: For each dataset, on the left is the probability that the part is the first part drawn (after the eye, which subjects are asked to draw first). On the right is the probability that conditioned on the row part being drawn, the column part is drawn next. In Creative Birds, we see that sketches typically start with either the beak (in case the initial stroke already provides part of the head), or the head. In both cases, the next part is often the body. From there either the wings or legs tend to be drawn. The tail tends to follow. The mouth is often drawn last. Similar patterns can be seen in Creative Creatures, except with larger variety due to the larger number of part options and larger diversity of creatures that subjects draw.

## F  IMPLEMENTATION DETAILS

**Training.** We found that the default learning rates used in the StyleGAN papers lead to over-optimization of the discriminator during the early training phase and prevent effective optimization of the generator on our datasets. Therefore, we did a grid search on the discriminator and generator learning rates in $\{10^{-3}, 5^{-3}, 10^{-4}, 5^{-4}\}$ and batch sizes in $\{8, 40, 200\}$, looking for training stability and the high generation quality. We picked a learning rate of $10^{-4}$ and a batch size of $40$ for both the discriminator and generator. We use the Adam optimizer Kingma & Ba (2014) following StyleGAN papers (Karras et al., 2019; 2020) with $\beta_1 = 0, \beta_2 = 0.99, \epsilon = 10^{-8}$. We multiply a trade-off factor equals to $0.01$ to the sparsity loss. As for the part selectors for both datasets, we train each on $80\%$ of the data with learning rate $2^{-4}$ and batch size $128$ for $100$ epochs when the training losses are observed to reach the plateau and a reasonable testing accuracy is achieved on the other $20\%$ data. Specifically, we get $63.58\%$ and $48.86\%$ part selection test accuracy on the Creative Birds and Creative Creatures datasets respectively. Our training time of each creature and bird part generator is approximately 4 and 2 days on a single NVIDIA Quadro GV100 Volta GPU.

**Data augmentation.** As discussed in Section 4, we apply a random affine transformation to the vector sketches as a data augmentation step in addition to the random horizontal flips during training.

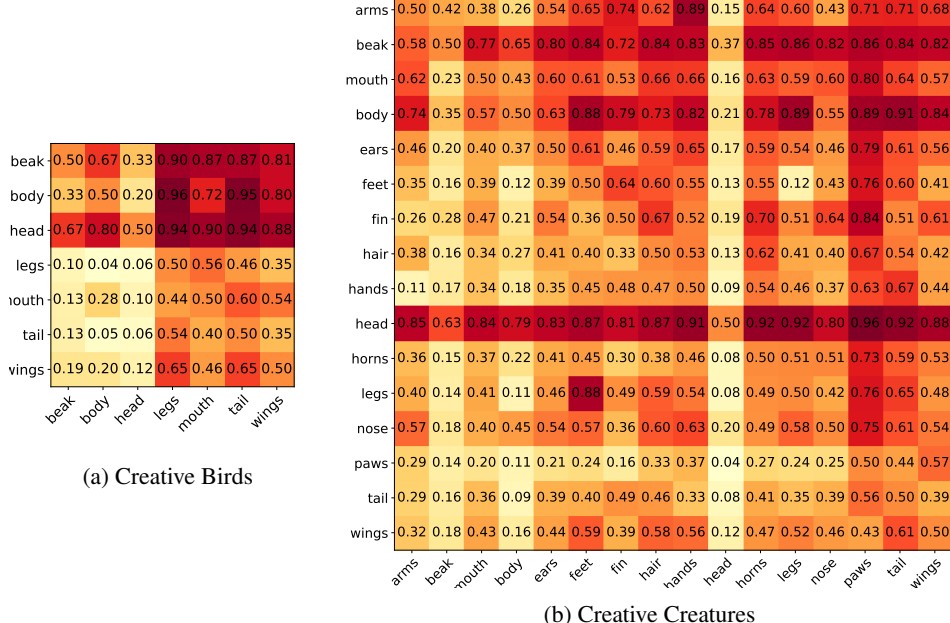

(a) Creative Birds

(b) Creative Creatures

Figure 8: Across all sketches that contain both parts, we show the probability that the part along the column was drawn (at some point) after the part along the row. In the Creative Birds dataset, the smaller parts such as the legs and mouth are often drawn after the larger parts including the head and body. In the Creative Creatures dataset, the head is often drawn before other parts. Hands, paws, feet, legs tend to be drawn later.

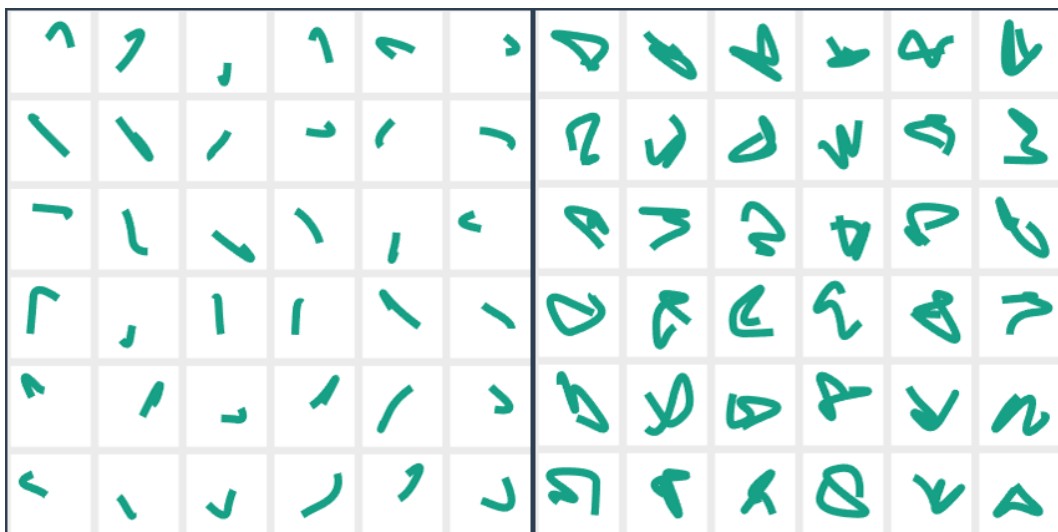

Figure 9: Example short (left) and long (right) random initial strokes used as a prompt in our data collection and as the first step in our sketch generation approach. The short strokes were used for Creative Birds and long for Creative Creatures. Thick strokes are shown for visibility.

Specifically, we used two sets of affine transformation parameters: 1) random rotation with angles $\theta \in \mathcal{U}[-15°, 15°]$, random scaling with ratio $s \in \mathcal{U}[0.9, 1.1]$, random translation with ratio $t \in \mathcal{U}[-0.01, 0.01]$, and a fixed stroke width $l = 2$ pixels; 2) $\theta \in \mathcal{U}[-45°, 45°]$, $s \in \mathcal{U}[0.75, 1.25]$, $t \in \mathcal{U}[-0.05, 0.05]$, and $l \in \mathcal{U}[0.5, 2.5]$ pixels. We found that a larger transformation and longer training time is especially useful when training part generators on the Creative Creatures dataset which is more challenging and parts often have fewer examples than Creative Birds. We also found

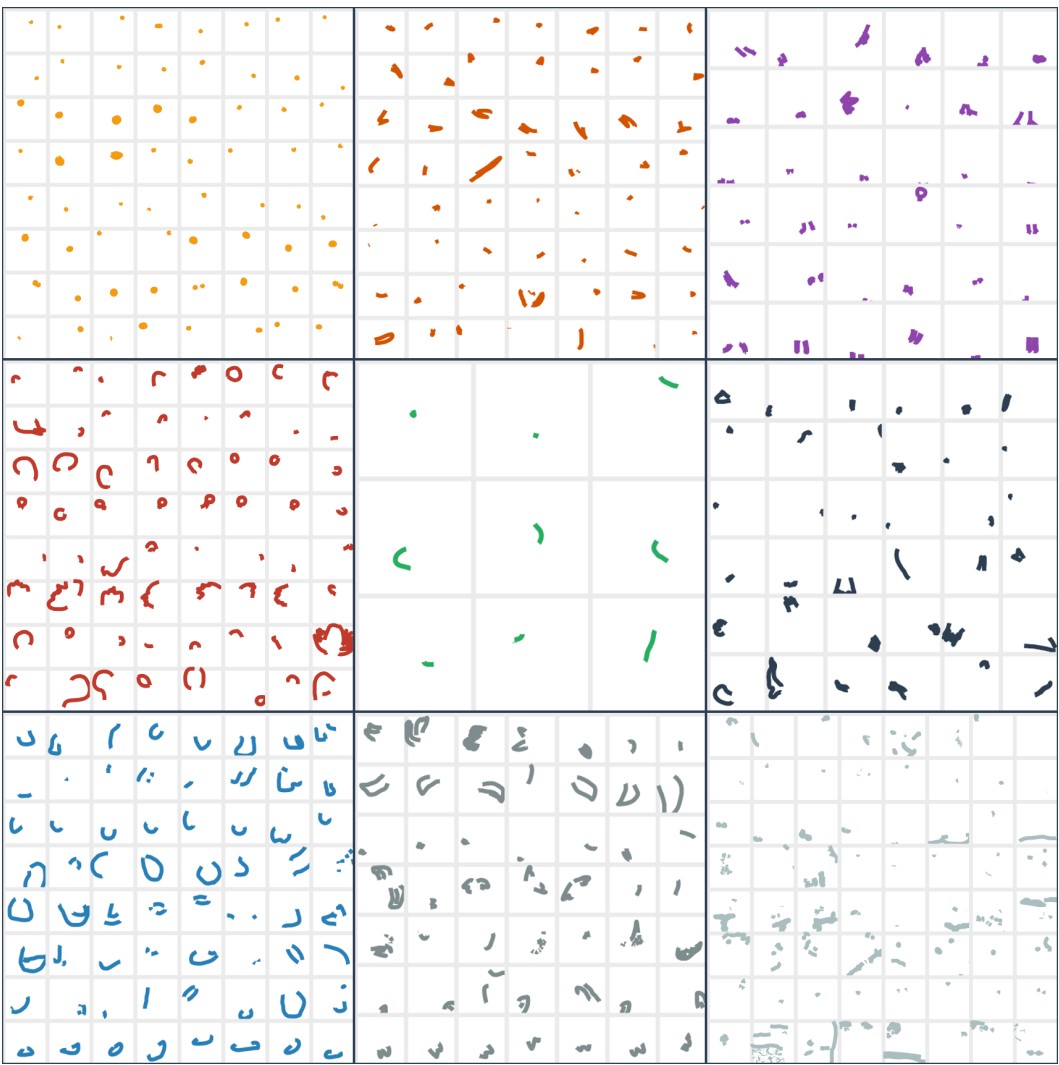

Figure 10: Random examples of parts from Creative Birds. Thick strokes are displayed for visibility. Top to bottom, left to right: eye, head, body, beak, mouth, wings, legs, tail, details. The number of examples in each block is roughly proportional to the percentage of sketches that have that part.

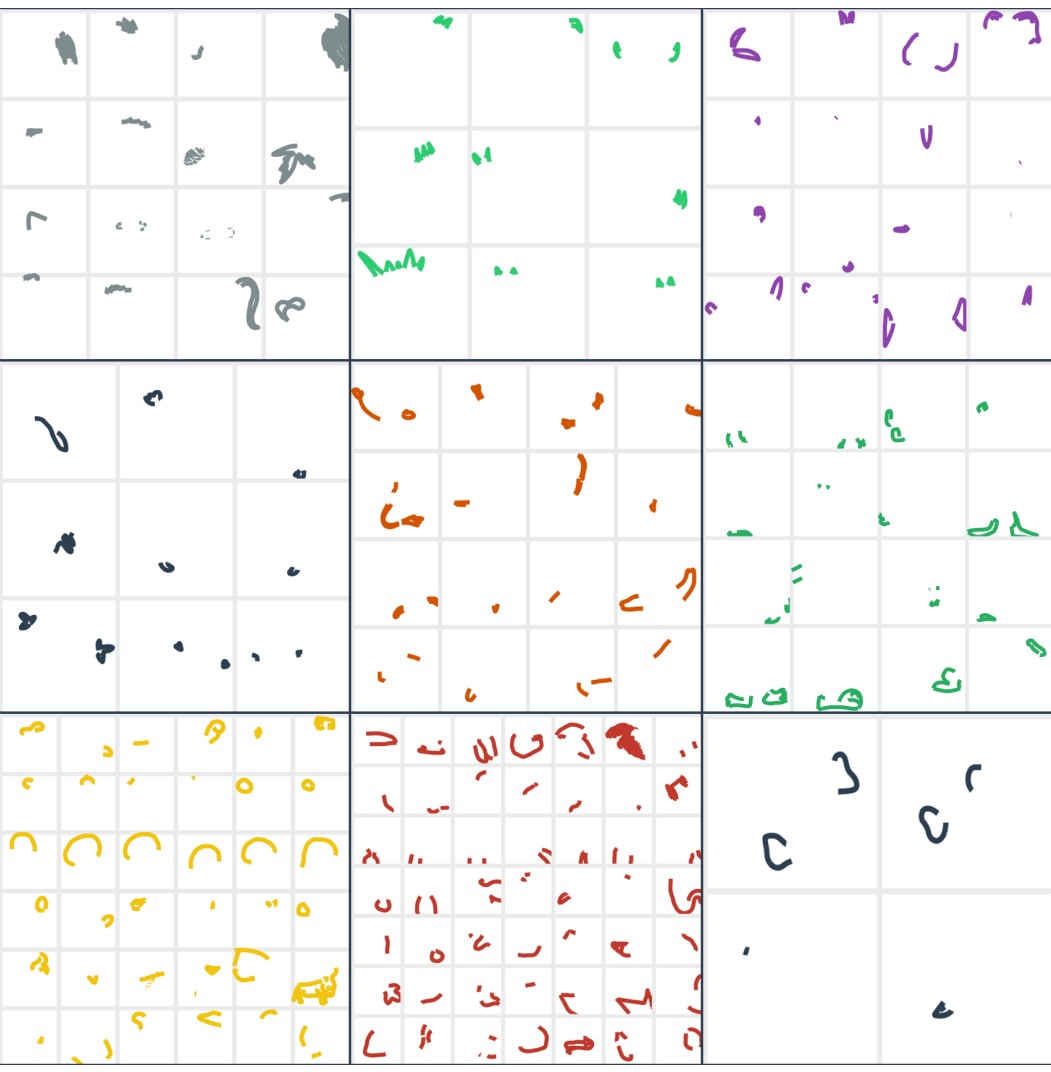

Figure 11: Random examples of parts from Creative Creatures. A subset of parts are shown, with preference for those not in common with Creature Birds. Thick strokes are displayed for visibility. Top to bottom, left to right: hair, hands, head, horns, arms, body, ears, feet, fin. The number of examples in each block is roughly proportional to the percentage of sketches that have that part.

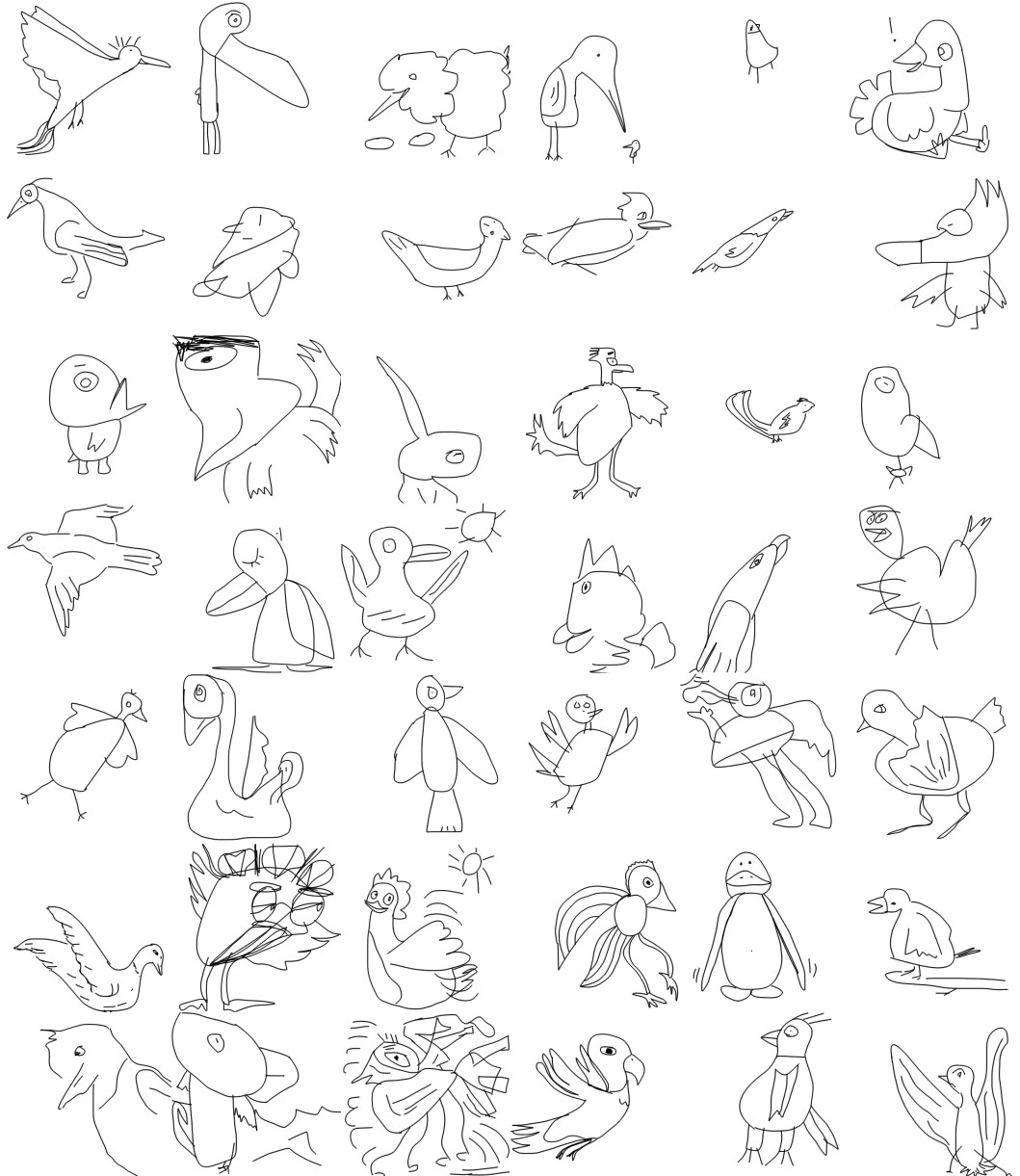

Figure 12: Random example sketches from our Creative Birds dataset. Entire sketches (including the "details") are shown.

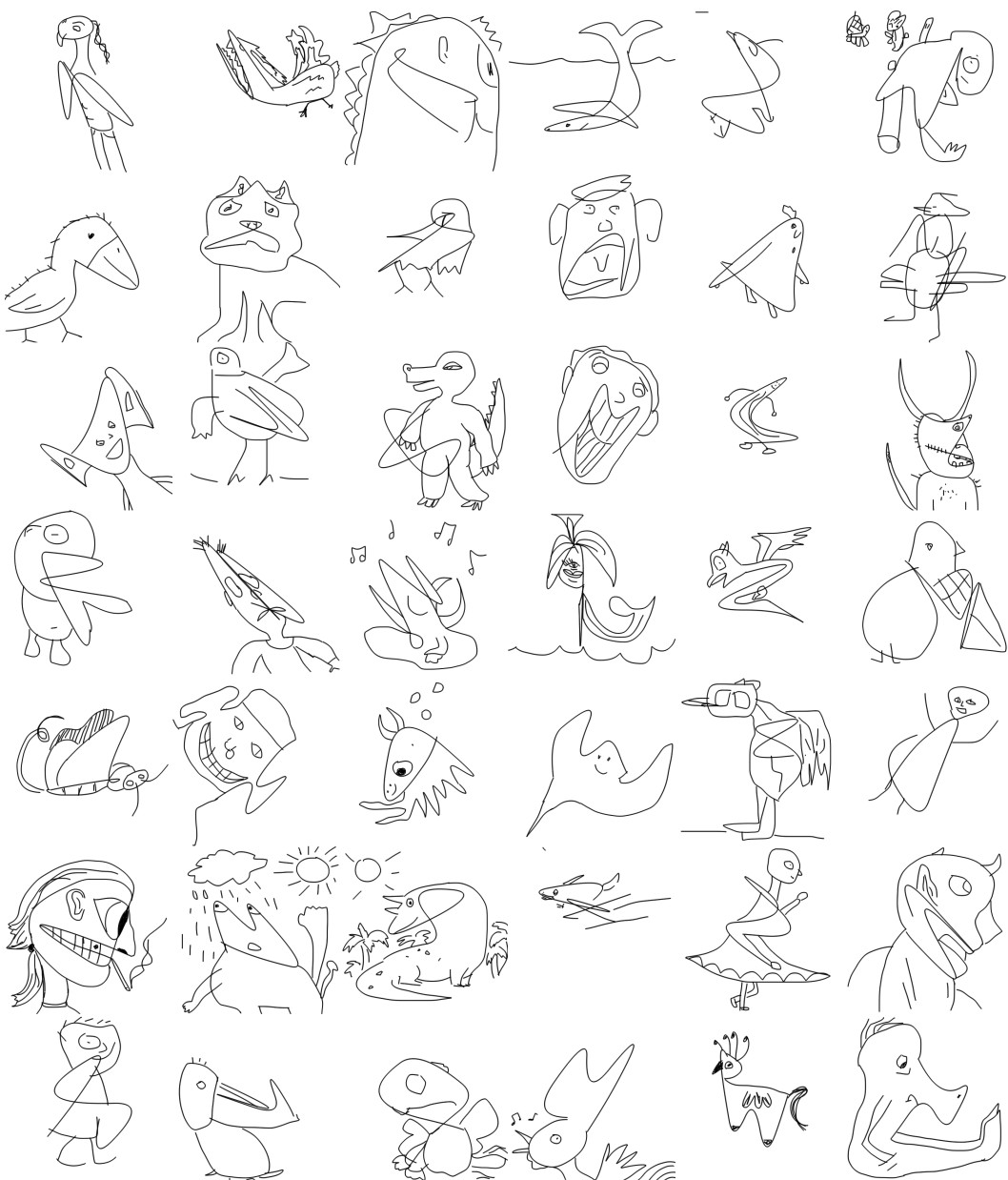

Figure 13: Random example sketches from our Creative Creatures dataset. Entire sketches (including the "details") are shown.

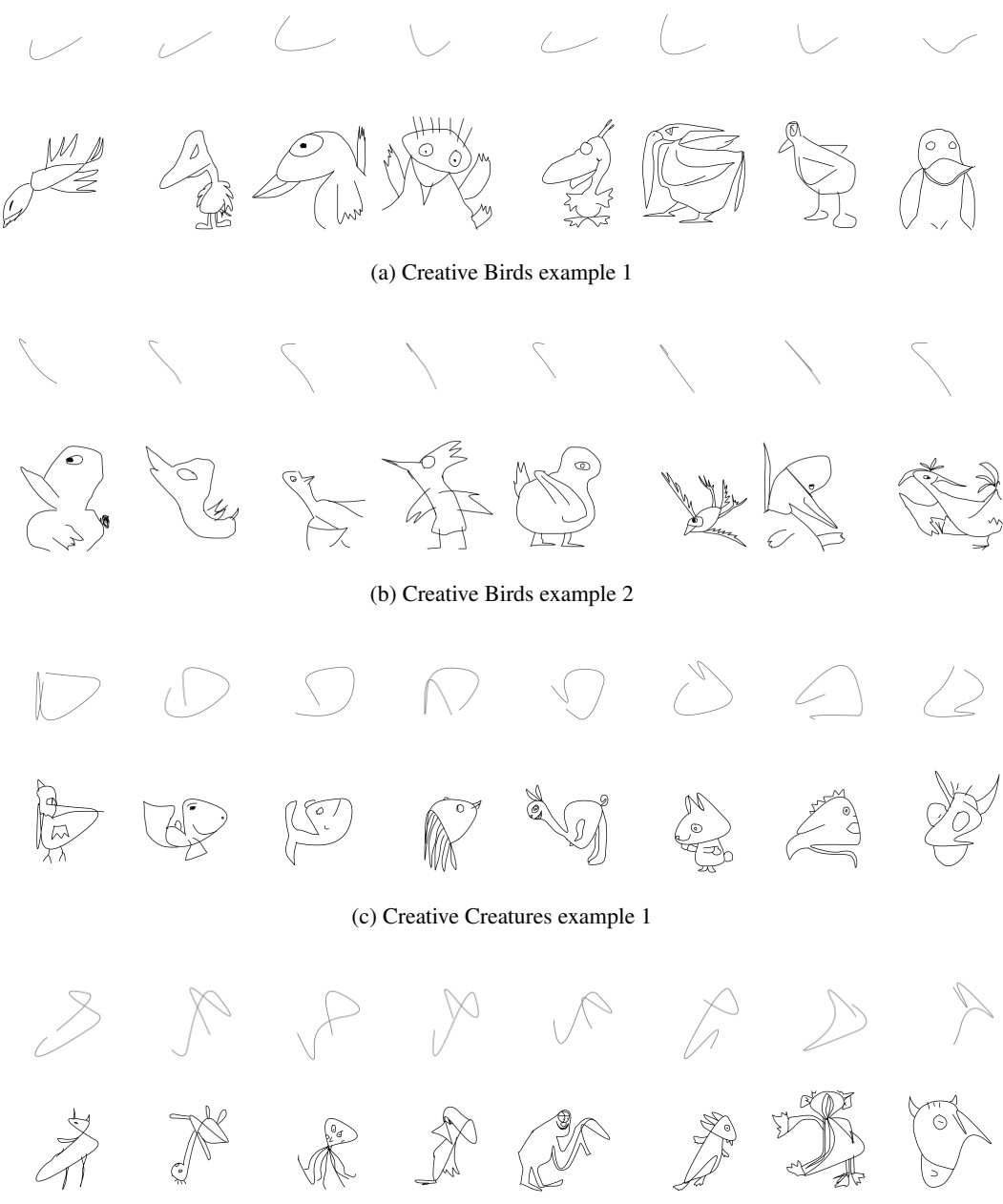

(a) Creative Birds example 1

(b) Creative Birds example 2

(c) Creative Creatures example 1

(d) Creative Creatures example 2

Figure 14: Sketches with similar initial strokes from Creative Birds and Creative Creature datasets. Similar initial strokes have diverse sketches that incorporate them. Note that it is much easier to find sketches with similar initial strokes in the Creative Birds dataset because the initial strokes are shorter. This also indicates that the Creative Creatures dataset is more challenging.

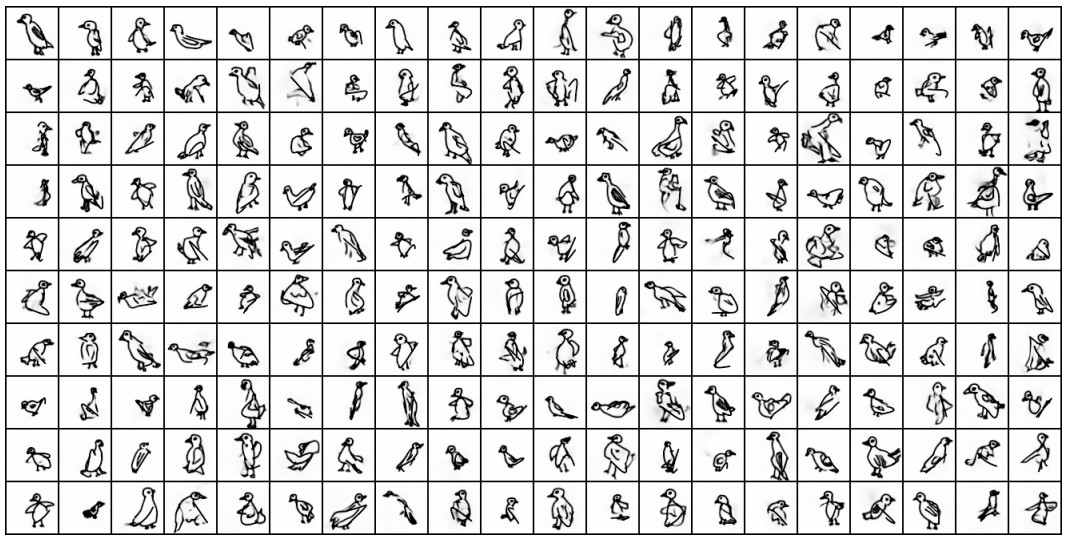

(a) Creative Birds.

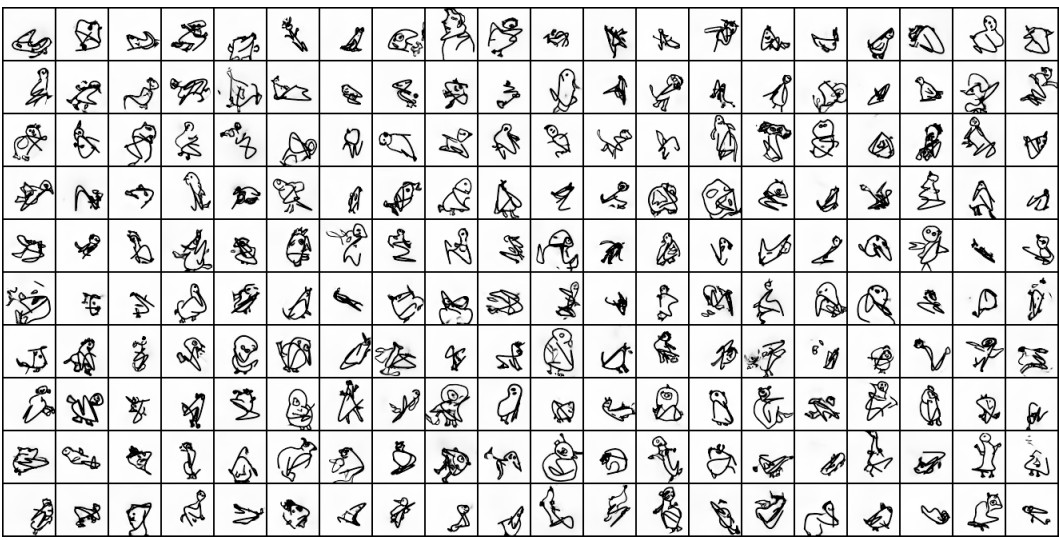

(b) Creative Creatures.

Figure 15: Uncurated creative sketches generated by DoodlerGAN.

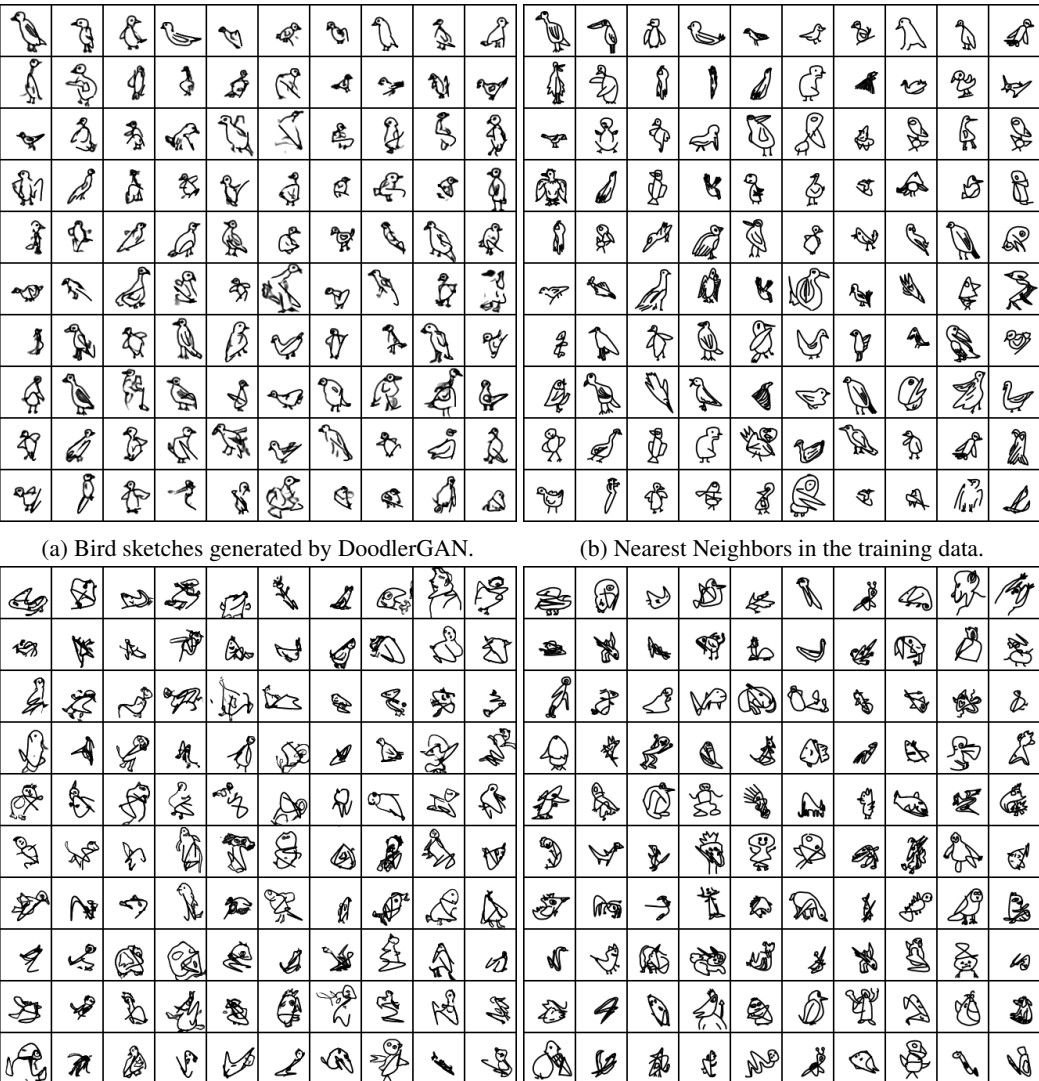

(a) Bird sketches generated by DoodlerGAN.

(b) Nearest Neighbors in the training data.

(c) Creature sketches generated by DoodlerGAN.

(d) Nearest Neighbors in the training data.

Figure 16: Nearest neighbors of the sketches generated by DoodlerGAN showing that the generations do not mimic the training data.

that the larger transformation also helps to increase the variation in the generation based on the noise but the general quality of generated parts suffers on the Creative Birds dataset. Consequently, we apply the larger augmentation when training the creature part generators and bird eye generator and smaller augmentation when training the other models. And we train the creature part generators for 60,000 steps and bird part generators for 30,000 steps.

**Inference.** For Creative Birds, we noticed that the quality of the generated sketch was higher when the generated eye was not too small and not too close to the initial stroke. Motivated by this observation, we used the following trick: We sample 10 eyes for each initial stroke and rank them based on the pixel sum and distance to the initial stroke. We combine the ranks using Borda count and pick the highest ranked eye for the following steps. Using this we see an improvement in the generation quality for Creative Birds, but not Creative Creatures.

Since humans in our data collection process were asked to add at least five parts to the sketch, we employ the same rule for our part selector – only after five parts have been added, the selector is given the option to predict stop. Once a part has been drawn, that part can not be selected again.

## G    DETAILS OF MODEL TRAINED TO COMPUTE FID AND DIVERSITY SCORES

FID computes the similarity between two image distributions. It first embeds the images into a feature space defined by a trained Inception model (Szegedy et al., 2016). It then fits a multivariate Gaussian to the feature vectors from both sets of images. Finally, it computes the Fréchet distance between the two Gaussians.

To this end, we trained an Inception model on the QuickDraw3.8M dataset (Xu et al., 2020). Specifically, we preprocess the dataset based on the instructions in the paper (Xu et al., 2020) and render each vector sketch into a $64 \times 64$ raster image. The dataset contains $345$ classes and each class contains $9,000$ training samples, $1,000$ validation samples, and $1,000$ test samples. We train the Inception model with the RMSprop optimizer (Tieleman & Hinton, 2017) for $50$ epochs. We use an initial learning rate of $0.045$ with an exponential decay at a rate $0.98$ after every epoch. The trained model achieved a reasonable test accuracy 76.42% ( state-of-the-art accuracy is 80.51% reported in SketchMate (Xu et al., 2020) using $224 \times 224$ images).

Generation diversity calculates the average pairwise Euclidean distance between the features calculated for sketches with a trained model to reflect the diversity of the generation. To be consistent, we use the same Inception model trained on QuickDraw3.8M to calculate these features.

## H    PROPOSED METRICS

Here we provide more details about the the characteristic score (CS) and semantic diversity score (SDS) metrics introduced in Section 5.1. We first construct the label sets $\mathcal{B}$ and $\mathcal{C}$ which include the labels from the $345$ QuickDraw classes that represent a bird or creature. The details of the two sets can be found below. Then for the Creative Birds and Creative Creatures datasets, we define the characteristic score as the percentage of the time a generated sketch is classified as a label in $\mathcal{B}$ and $\mathcal{C}$ respectively. The characteristic score gives us a basic understanding of the generation quality, indicating whether they are recognizable as the relevant concepts (birds and creatures) we used to create the dataset. We define the semantic diversity score as follows:

$$\text{SDS} = p_\mathcal{C} \sum_{l \in \mathcal{C}} -\frac{p_l}{p_\mathcal{C}} \log \frac{p_l}{p_\mathcal{C}} = -\sum_{l \in \mathcal{C}} p_l \log \frac{p_l}{p_\mathcal{C}},$$

where $p_l$ represents the average probability that a sketch in the generation collection is classified as label $l$, and $p_\mathcal{C}$ represents the average probability that a generated sketch is classified as a creature i.e., $p_\mathcal{C} = \sum_{l \in \mathcal{C}} p_l$. The intuition is similar to Inception Score (IS) (Salimans et al., 2016). Except that SDS does not penalize a sketch for receiving a high probability for more than one label. In fact, a sketch that looks like a combination of two creatures is probably quite creative! Such examples are shown in Figure 6 in the paper. Thus, in SDS we only look at the sum of the label distribution of the generation collection to capture two aspects of creativity: Across generations 1) the distribution should have a high variety across creature categories (measured by the entropy of the marginal creature distribution) and 2) the distribution should have the most mass on creature categories.

The bird label set $\mathcal{B}$ and creature label set $\mathcal{C}$ from the QuickDraw categories are as follows:

$\mathcal{B} = [bird, duck, flamingo, parrot]$

$\mathcal{C} = [ant, bear, bee, bird, butterfly, camel, cat, cow, crab, crocodile, dog, dolphin, duck, elephant, fish, flamingo, frog, giraffe, hedgehog, horse, kangaroo, lion, lobster, monkey, mosquito, mouse, octopus, owl, panda, parrot, penguin, pig, rabbit, raccoon, rhinoceros, scorpion, sea turtle, shark, sheep, snail, snake, spider, squirrel, swan, tiger, whale, zebra]$

## I    MULTIMODAL PART GENERATION

Conditional GANs tend to ignore the input noise and base their generations completely on the conditional information (Isola et al., 2017). We encounter similar challenges in our approach, especially for the parts later in the order when the conditional information is strong. For the eye generator, when the conditional signal is weak (just the initial stroke), the model responds to the noise well. We demonstrate this with the style mixing experiment conducted in the StyleGAN paper (Karras

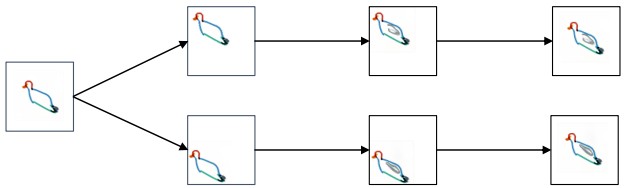

Random translation    Part generation   Reverse translation

Figure 17: Conditioning perturbation trick illustration.

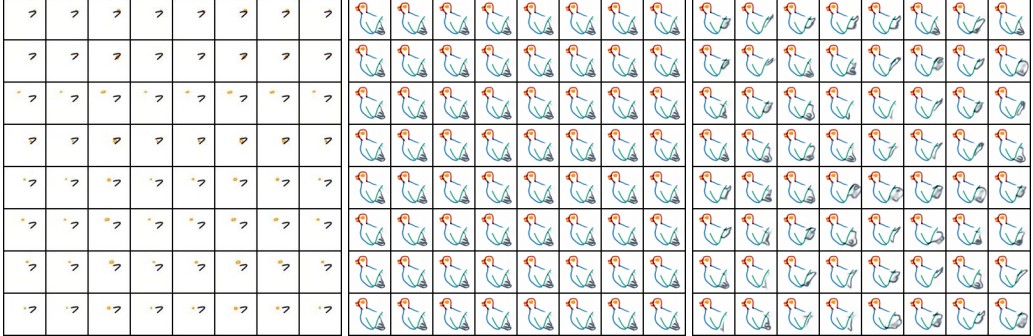

(a) Bird eye generations with mixed styles.

(b) Bird tail generation without conditioning perturbation.

(c) Bird tail generation with conditioning perturbation.

Figure 18: We show various parts generated by our model when conditioned on different noise vectors. The eye generator naturally responds to the noise – we see intuitive style mixing results in (a). For the other parts, we show that diverse parts can be generated (different appearances and positions) using our conditioning perturbation trick. See text for details.

et al., 2019). Specifically, given the same initial strokes, we sample two random noise vectors and feed them into different layers of the generator. In Figure 18a, each row has the same input noise for the lower generator layers and each column has the input same noise for the higher generator layers. We see that the lower-layer noise controls the position of the generated eyes while the higher-layer controls the size and shape of the eyes.

However, for generators of later parts, the generation tends to have little variation given different noise vectors as illustrated by the tail generation results in Figure 18b. Although the generator does not respond to the noise, we find that it is very sensitive to the conditional information. Specifically, if we slightly translate the input partial sketch, the generator generates a part with different appearance and positions. As shown in the Figure 17, we randomly translate the input partial sketch, feed them to the generator, and then translate the generated parts back to the correct positions. Figure 18c shows different generations for the tail given the same input partial sketch. We call this trick **conditioning perturbation**. This trick is motivated by the formulation in SketchRNN (Ha & Eck, 2018) where GMM parameters instead of deterministic points are used as the RNN output as a mechanism to introduce variation.

## J    GENERATIVE V.S. RETRIEVAL

In addition to the generation-based methods (StyleGAN2, SketchRNN), that we compared to in the main paper, we now consider a strong retrieval-based baseline. Specifically, we hold $5\%$ of the sketches in our datasets to use as query sketches. We extract the first $N$ parts from these sketches and use them to match against partial sketches containing $N$ parts from the remaining 95% of the dataset using the average Chamfer distance across parts. The rest of the matched sketch is returned to complete the query partial sketch.

A limitation of this approach is that the quality of the match will deteriorate if the application requires multiple completions of the same query partial sketch (e.g., to present as options to a human-in-the-loop). A retrieval approach will have to retrieve farther neighbors to find more completions. A generative approach can simply generate multiple samples as completions, with no systematic degradation in quality across samples as shown in Figure 18c.

We evaluate this via human studies, we hypothesize a scenario where 32 candidate completions are desired. We compare the 32nd retrieved completion with a "32nd" sample from DoodlerGAN (in our approach there is no natural ordering to the generations), and ask human subjects in which sketch the query is better integrated. We experiment with 200 queries containing 2 and 3 parts each. DoodlerGAN is statistically significantly better than the retrieval approach, preferred 54% and 55% of the times with 2- and 3-part queries respectively.

Recall that the retrieval approach, by definition, reproduces human sketches as completions. On the other hand, we saw in Figure 16 that DoodlerGAN generates sketches that are different from training data, making it conceptually a more creative approach.

## K  ABLATION STUDY

These ablation studies demonstrate the effectiveness of different design choices in DoodlerGAN.

**Part generator:** Figure 19a shows sketches generated if the entire partial sketch is used as the conditioning information without the part-wise channel representation. We see that it is difficult for the part generator to learn the spatial relationships between different parts. Notice that the head (red part) often does not include the eye inside it. Out of a sample of 512 head generations, we find that 57.4% of the time the generated head does not surround the eye when not using part channels, compared to 88.7% of the time with part channels. Figure 19b shows generations without the U-Net architecture, directly concatenating the encoder output with the noise vector instead. We see that the quality of the part composition is much worse than when using the U-Net (Figure 19c) which encodes the spatial information of the conditioning sketch in the feature map.

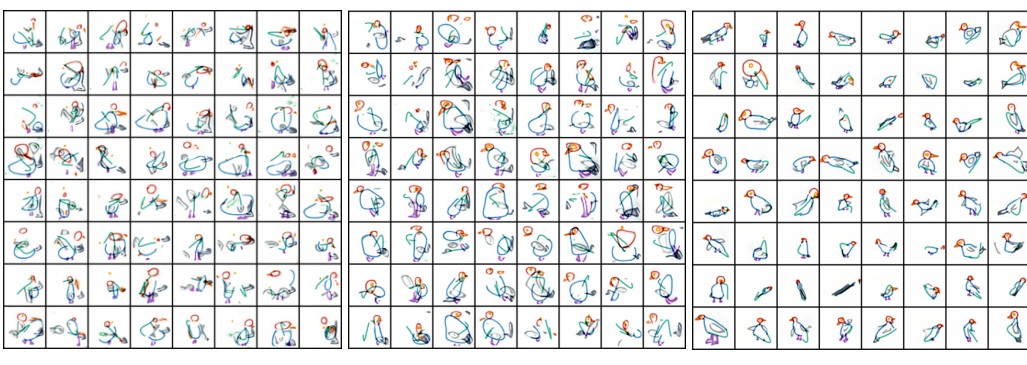

(a) Without part channels.  (b) Without U-Net connections.  (c) Full model.

Figure 19: Ablation studies on the part generator.

**Part selector:** Figure 20 shows sketches generated if instead of using a part selector, we generate parts in a fixed order (the most common order in the dataset). We find that the use of a part selector allows the model to use the initial stroke much more effectively. Consider the two sketches with red boundaries shown in Figure 20a. They have the same initial strokes. When using a part selector, a head is not generated and the initial stroke is used as the head. Whereas the generation with a fixed order still generates the head and this error propagates to subsequent part generations. Similar observation can be made for beak generation in the two sketches with blue boundaries shown in Figure 20a. Furthermore, Part selector plays an important role in choosing appropriate part lists and stopping the generation in an appropriate step. For instance, it is not reasonable to generate all 16 part in every creature sketch, which greatly undermines the quality as shown in Figure 20b. A part selector is also important if one wants to implement our method in a human-in-the-loop scenario

where the model would need to identify what the human has already drawn, and then decide what to draw next.

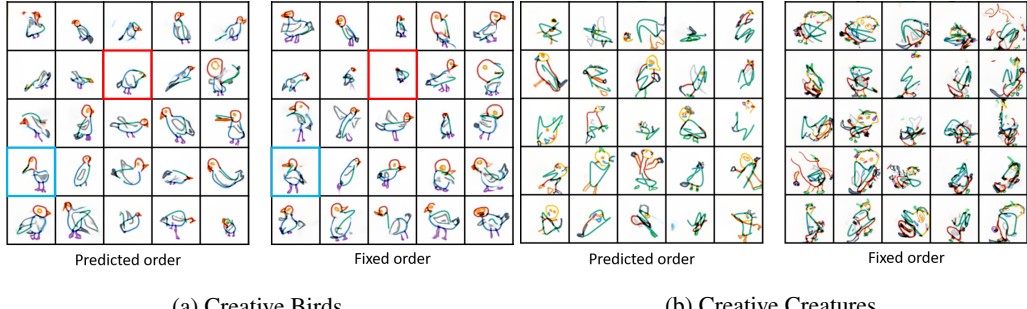

(a) Creative Birds.                    (b) Creative Creatures.

Figure 20: Ablation study on the part selector.

## L  CONVERTING GENERATED SKETCHES TO VECTOR IMAGES

DoodlerGAN generates sketches in raster image format. One disadvantage of raster images is that their resolution can not be easily increased without introducing artifacts. The benefit of vector images is that they can be rendered at arbitrary resolutions. We convert raster images of sketches to vector representations (SVGs) using the open source tool `mkbitmap`, which relies on the `potrace` program ((Selinger, 2003)) based on a polygon tracing algorithm. The parameters used are shown in Table 4. Examples conversions are shown in Figure 21. Notice that in addition to the change in format, the conversion also introduces a different aesthetic to the sketch.

We conduct a human study to evaluate this aesthetic. We found that subjects prefer the new aesthetic 96% of the times! Note that this is while keeping the resolution of the two aesthetics the same. We hypothesize that this may be because the converted images do not have any blurry artifacts and the strokes are smoother. For completeness, in Figure 21 we also show a couple of these converted sketches at a higher resolution.

Table 4: `mkbitmap` & `potrace` parameters

| Parameters | threshold grey value | bandpass filter radius | scale | interpolation | backend | path type |
|---|---|---|---|---|---|---|
| **values** | 0.2 | 4 | 2 | cubic | svg | group |

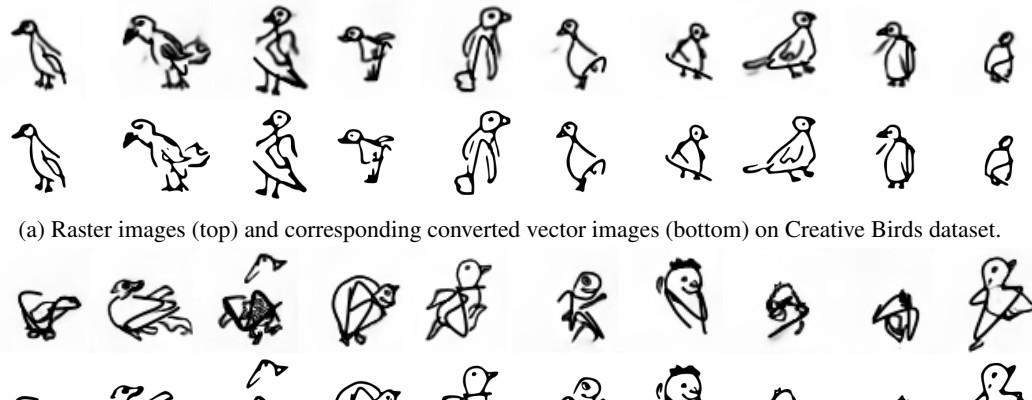

(a) Raster images (top) and corresponding converted vector images (bottom) on Creative Birds dataset.

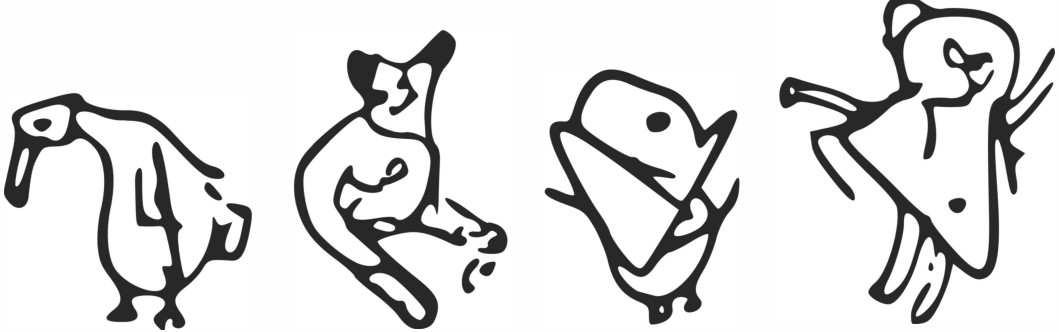

(b) Raster images (top) and corresponding converted vector images (bottom) on Creative Creatures dataset.

(c) Cherry-picked generations from DoodlerGAN converted to a vectorized representation and displayed at higher resolution.

Figure 21: Converting DoodlerGAN's sketches from raster images to vector representations

