# OpenReview forum: "Creative Sketch Generation"
_ICLR.cc/2021/Conference — ICLR 2021 Poster_

### Official Review · AnonReviewer2 · 2020-10-14
**An improved approach to automatic sketching**

**Rating:** 7
**Confidence:** 3

**Review:**

The contribution of this paper is twofold. First, the authors introduce a pair of manually collected (via AMT) datasets of creative sketches (birds and general creatures) each along with part annotations. Second, they propose a part-based Generative Adversarial Network for the generation of unseen compositions/configurations of novel parts (legs, body, etc.) for creative sketches. The latter is quantitatively and qualitatively (via human studies) evaluated.

Strengths

- Generally, the paper is well introduced and easily understood. Especially, the provided dataset is very useful in promoting the development of the problem addressed: creative sketch generation. Although not being a really “novel” idea/data, providing new benchmarks/datasets/competitions for the AI community is always refreshing.

- The authors also include a generative architecture which is then quantitatively evaluated against several working implementations of the pretrained SOTA baselines. They also provide easy-to-understand evaluation metrics allowing the community to keep on working on the task at hand. The experimental results demonstrated the effectiveness of the proposed method.

- The illustrative experimental setting shows the usefulness of the datasets, the potential of their generative approach and its easiness of use.

Weaknesses

- Of course, this is a general concern any time a diagnostic dataset/tool is introduced. Rarely is an immediately surprising insight offered in the same paper. The value of such datasets/tools is often clear only in hindsight with the benefit of time. If they are useful, they see organic adoption. If they are not, they organically fall to the wayside.

- Furthermore, one can argue that there is no particularly novel insights in this paper (no science behind the collection of the datasets and the proposed GAN is a logic incremental evolution of the SOTA). However, the paper is very well written and structured, the methodology followed is correct and the experimental setting is comprehensive (with promising results), so I'm happy to let the noisy process of science (and reviewing process) figure out the value here.

---

> ### Author Response · Authors · 2020-11-18
> **Response to Reviewer 2**
>
> Thank you for your time and feedback!
>
> ***“Of course, this is a general concern any time a diagnostic dataset/tool is introduced. The value of such datasets/tools is often clear only in hindsight with the benefit of time.”***
>
> Agreed. Note that our part-based generation framework was already inspired by this dataset :) Given that these are the first two large scale sketch datasets where semantic part annotations and text captions are available, as R4 also suggested, other tasks such as fine-grained domain adaptation, text-to-sketch generation, creative sketch captioning, sketch segmentation can be studied (in addition to creative sketch generation explored in this paper).
>
> ***“... the proposed GAN is a logic incremental evolution of the SOTA.”***
>
> Note that direct application of a SOTA conditional GAN model works quite poorly compared to our approach (as demonstrated through human studies and quantitative evaluation). Additionally, incorporating a part-based generation process was not trivial. For instance, including part-based channels for both the generator and discriminator made the discriminator too strong, leading to not enough signal for the generator to train. We thus incorporated a sparsity loss to give the generator additional signal. This is one example -- our paper details many of the other design choices that were necessary to make the proposed approach work effectively. See Appendix K for more ablation studies of our model.

---

### Official Review · AnonReviewer3 · 2020-10-21
**Delightful, well written paper! I have concerns about its fit here.**

**Rating:** 7
**Confidence:** 4

**Review:**

In this paper, the authors collect a large dataset of sketches of imaginative birds and other creatures, then train a GAN to produce similarly creative sketches. They use both quantitative and qualitative methods to evaluate their work and find that the generated sketches are of higher quality than both other models and human sketches, and are novel.

This topic is delightful and your paper is quite well written. My only concern is fit for the venue. This isn't a paper about a learned representation and doesn't encounter any theoretical issues regarding representations, why should it be published at ICLR rather than another venue, e.g. ICCC (https://computationalcreativity.net/iccc21/)? There are plenty of papers at ICLR that aren't about representation learning these days, so I'm still inclined to accept, but I'd like to hear an argument.

More specifically, I appreciate the level of detail regarding the dataset collection study and your model training process. I would also like to see more discussion of your "conditioning perturbation" trick. Why does it work? Can you characterize the effect the perturbations have on the generated sketches?

I'm a little confused about Figure 5: were the images from each method compared against the same set of images? If not, you may need to do some more sophisticated statistical analysis than the Bernoulli confidence interval to judge the probability that each method is chosen above each competitor. I also find it hard to believe that the human drawings perform significantly worse than the GAN drawings across the board. Does that indicate a problem with the dataset you constructed or some bias among your participants?

Very minor issue, but I think you also have a typo in Table 1, the DS column should probably be labeled GD to match the table caption.

---

> ### Author Response · Authors · 2020-11-18
> **Response to Reviewer 3**
>
> Thank you for your time and feedback.
>
> ***“My only concern is fit for the venue. This isn't a paper about a learned representation… There are plenty of papers at ICLR that aren't about representation learning these days, so I'm still inclined to accept, but I'd like to hear an argument.”***
>
> We do not use any hand-designed features, so our model does need to learn a good representation of sketches to support the generation process.
>
> Past publications at ICLR have included papers on creative applications such as image editing [1, 5], sketch generation [2], image-to-image translation [3], neural painting [4], music translation [6] and music generation [7]. Reviewer 4 said "the topic of creative generation is interesting and gradually becomes the future research direction in the image generation field".
>
> [1] Brock et al. "Neural Photo Editing With Introspective Adversarial Networks." ICLR 2017.
> [2] Ha et al. "A Neural Representation of Sketch Drawings." ICLR 2018.
> [3] Ma et al. "Exemplar Guided Unsupervised Image-to-Image Translation with Semantic Consistency." ICLR 2019.
> [4] Zheng et al. "Strokenet: A neural painting environment." ICLR 2019.
> [5] Mo et al. "InstaGAN: Instance-aware Image-to-Image Translation." ICLR 2019.
> [6] Mor et al. "A Universal Music Translation Network." ICLR 2019.
> [7] Hawthorne et al. "Enabling Factorized Piano Music Modeling and Generation with the MAESTRO Dataset." ICLR 2019.
>
> ***About the "conditioning perturbation" trick***
>
> Thanks for the questions. We added Figure 17 to the Appendix. We see that as we randomly translate the input sketch, we can get fairly different generated sketches. More results on tail generation can be found in Figure 18, where using conditioning perturbation in Figure 18 (c) results in much more variation than without conditioning perturbation in Figure 18 (b).
>
> To be honest, it is not entirely clear to us why it works. There may be connections to adversarial examples where tiny perturbations on the input can lead to dramatic changes in the model behavior. As far as we know, there are a few works that study adversarial examples in (unconditional) generative models [1] but not many in conditional generative models. Studying this further is open for future work.
>
> [1] Kos et al. "Adversarial examples for generative models." IEEE Security and Privacy Workshops, 2018.
>
> ***“were the images from each method compared against the same set of images?”***
>
> Yes, the images are aligned. Note that this notion of alignment is meaningful for conditional approaches (images being compared use the same random initial stroke). There is no natural notion of alignment for the unconditional approaches. The conditional approaches form the stronger and more natural baselines.
>
> ***“I also find it hard to believe that the human drawings perform significantly worse than the GAN drawings across the board. Does that indicate a problem with the dataset you constructed or some bias among your participants?”***
>
> It is unlikely that’s the case because we see this only on the birds dataset (not on creatures), and even on birds, only for the two metrics “look more like birds?” and “the worker likes better?”. We don’t see this across the board. There is a *significant* gap between DoodlerGAN generations and human sketches on the Creative Creatures dataset (which is more diverse and challenging).
>
> Note that the same data collection protocol was used for Creative Birds and Creative Creatures. Similarly, the same interface was used for the human evaluation on both datasets and all metrics.
>
> 178 unique individuals from AMT participated in our studies. For this large a group, it seems unlikely that they are biased towards DoodlerGAN generations in some way.
>
> Finally, on birds and “looks more like birds?”: Our quantitative evaluation also shows that DoodlerGAN generations have a higher chance to be classified as a bird than human drawings with a classifier trained on QuickDraw.
>
> Note to make sure there is no confusion when interpreting Figure 5: In Figure 5, the evaluation metric “Human drawn” on the X-axis should not be confused with “Creative Datasets” in the legend. “Human drawn” shows how often subjects thought the sketch was drawn by a human. Most bars being above the top dashed line means sketches generated by DoodlerGAN were considered to be more likely to be drawn by humans than the baselines we compare against. “Creative Datasets” in the legend corresponds to the sketches that were drawn by humans (as opposed to being generated by a baseline). Most bars corresponding to “Creative Datasets” do not cross the top dashed line, and are significantly lower than the lower dashed line for the Creative Creatures dataset -- indicating that human sketches are preferred over DoodlerGAN generations.

---

> > ### Comment · AnonReviewer3 · 2020-11-24
> > **Response**
> >
> > Thank you for your thoughtful reply! I think your responses are satisfactory on the issues I raised. I'm increasing my score accordingly.
> >
> > I realize now I completely misread Figure 5: each bar represents your model's performance against the labeled model, rather than the labeled model's performance. While the figure caption makes sense in hindsight, it was not clear for the few minutes of looking at it. A different visualization, maybe with the area between the bar and the top of the plot filled in a different color, might be clearer.

---

### Official Review · AnonReviewer4 · 2020-10-28
**A good paper**

**Rating:** 7
**Confidence:** 4

**Review:**

This paper put forward two Creative datasets - Creative Birds and Creative Creatures, which have corresponding part annotations. Based on Part selector and Part generator. The authors demonstrate a multi-stage sketch generation approach - DoodlerGAN, and achieve the STOA compared with other methods in several metrics.

The topic of creative generation is interesting and gradually becomes the future research direction in the image generation field. There are some pros:
- The datasets provide a novel perspective to decouple the basic elements of sketch images with fine edge semantic labels and text descriptions. These annotations could be used in the fine-grained domain adaption task, text-to-sketch generation or some reverse tasks. The reviewer thinks this contribution is meaningful and solid.
- The sketch generation fashion is enlightening. Although the idea of multi-stage generation has been used in many synthesis tasks, it's mainly for different resolutions or granularities (local and global). The partial strokes combine the overall sketch images is a domain-specific technique. The authors reimplement in deep learning way. The reviewer has to say that it may have some limitations on other fields, it's appreciated if the authors can share new possibilities, but that's not the point.
- The experiments are solid. Quantitative evaluation and human evaluation both outperform other baseline methods. It's convincing.

Some concerns are also proposed:
- How are the predicted labels added to the generator? It doesn't seem to show up on Figure 3(a). The reviewer wants to know more details.
- The reviewer has some confusion about the noise range in Figure 3(b), which is N(0,1). However, there is different descriptions in sec 4.2 - N(0,2). Which one is the true sampling distribution?
- The full loss functions are lack in the mainly body, which should be depicted more clearly.

---

> ### Author Response · Authors · 2020-11-18
> **Response to Reviewer 4**
>
> Thank you for your time and feedback.
>
> ***“How are the predicted labels added to the generator?”***
>
> There is a different generator for each part. During inference, we use labels produced by the part-selector as the indicator to select the corresponding part generator.
>
> A follow-up question might be: would a unified generator work better? We did not find this to be the case in our experiments. We tried a single generator for all the parts and input the labels as a one-hot vector to the generator and discriminator in multiple ways -- concatenating it with the noise vector, using it to select part-specific starting feature maps in the generator, etc. We also tried to train the part selector and generators in an end-to-end fashion. We found noticeably worse generation quality compared to using different generators for different parts. However, we think that a unified generator would likely generalize better, especially when some parts have fewer training examples. It is also a more elegant design. It would be interesting to explore this further in future work.
>
> ***“The reviewer has some confusion about the noise range in Figure 3(b), which is N(0,1). However, there is different descriptions in sec 4.2 - N(0,2). Which one is the true sampling distribution?”***
>
> These are describing two different distributions.
>
> - In Figure 3 (b), N(0,1) is the prior distribution of the noise used in the GAN model.
>
> - In Sec 4.2, N(0,2) is used to sample the parameters in our proposed conditioning perturbation. Specifically, as shown in Figure 17, we randomly translate the input partial sketch with distance sampled from N(0,2), feed the sketch to the generator, and then translate the generated parts back to the correct positions. This improves the diversity of generations given the same input partial sketch.
>
> ***“The full loss functions are lack in the mainly body, which should be depicted more clearly.”***
>
> All losses except for the sparsity loss are borrowed from [1]. We did not replicate those for space considerations, but will reconsider for the final version of the paper.
>
> [1] Tero Karras, Samuli Laine, Miika Aittala, Janne Hellsten, Jaakko Lehtinen, Timo Aila. "Analyzing and improving the image quality of stylegan." CVPR 2020.

---

> > ### Comment · AnonReviewer4 · 2020-11-24
> > **The paper can be accepted**
> >
> > Thanks for the response. I recommend to accept this paper.

---

### Official Review · AnonReviewer1 · 2020-10-28
**Introduces two novel doodle sketch datasets as well as a generative model for producing new sketches but does not seem to provide a substantial advantage over existing work**

**Rating:** 6
**Confidence:** 4

**Review:**

### Summary
This paper introduces two creative sketch datasets of birds and creatures, segmented into parts, each with ~10k doodles collected from Amazon MTurk workers. In a user study, people tend to favor sketches from their dataset over the similar Gogole QuickDraw sketches. Additionally, the authors propose a GAN architecture for generating novel sketches in an incremental fashion, one part at a time. They provide many qualitative results as well as human studies to validate their approach.

### Explanation of Rating
While the new datasets look nice, I'm not sure that they sufficiently different from or better than existing sketch datasets. Overall, the scale, complexity, and diversity of the sketches are roughly the same as some of the other datasets mentioned in the paper. It isn't obvious to me which novel applications or approaches are made possible with this dataset that were not possible before.
With respect to the proposed generative model, I think a user-in-the-loop interface is a reasonable approach, but though the model seems compatible with such a system, the authors do not actually implement it. It would be nice to see this interface in practice, since without it, the results do look a bit better but are not particularly more useful than those from other GAN models.

### Pros
- The UI and methodology for data collection, in which a user is asked to first draw an eye, followed by semantic parts of their choosing is interesting and appears to be conducive to better sketch quality.
- The paper is well written and contains many qualitative results and figures.
- The labeling of sketches by semantic parts presents an advantage over previous datasets.
- The iterative/incremental design of the generative model is a nice step towards computer-aided user-in-the-loop creative design.

### Cons
- While the authors claim that Figure 2 shows that their dataset is more diverse and creative, I'm not sure if that's objectively the case and if the marginal improvement is sufficiently important for downstream tasks.
- I understand the authors' argument that part-level granularity is more reasonable than stroke-based granularity, but I think that one major insight that can be gained from studying sketch datasets is with respect to how humans draw. For this reason, it would be useful to have information about the order of individual strokes, which the dataset does not appear to include.
- I'm not sure about the argument that raster images are a better format for a generative model than vector. It seems like the long-range relationships and connectivity as well as the sparsity inherent in a vector representation are pretty important in a sketch. Consequently, the some of the results produced by the model appear to suffer from topological artifacts and poorly-defined line segments.
- It would be useful to provide a citation for the doodling process used for data collection.
- How were the 100 sketches from Creative Birds and QuickDraw chosen for the user study?
- The paper claims the importance of certain architecture choices, e..g, that part channels help the model better predict the part locations. Some ablation study to verify this would be appropriate.
- Page 4: "subjetcs" (typo)

---

Thank you to the authors for their comments and clarifications.  I still am skeptical that the proposed dataset is a fundamental improvement over Quick Draw---ultimately, both datasets contain compact sketches from a single category containing relatively few strokes. But given that your updates and clarifications have addressed many of my questions, I am raising my score.

---

> ### Author Response · Authors · 2020-11-18
> **Response to Reviewer 1**
>
> Thank you for your time and feedback.
>
> ***“While the new datasets look nice, I'm not sure that they sufficiently different from or better than existing sketch datasets”***
>
> There are two key features of our datasets that differentiate them from existing ones:
>
> 1.  Subjects drawing the sketches were not tasked with creating realistic sketches. The data collection protocol explicitly set the task up as a creative task. As a result, the sketches in our datasets are more creative, often depicting fictional birds and creatures, not found in existing datasets.
>
> 2.  Our datasets are the first large scale sketch datasets (10k sketches each) where **semantic part annotations** and **free-form text captions** are available. These can be useful for a variety of tasks (some also suggested by R4) including sketch segmentation, fine-grained (part-based) image-sketch translation or domain adaptation or sketch based image retrieval, text-to-sketch generation, sketch captioning, etc. As the research community explores these datasets, novel tasks and modeling challenges will likely emerge.
>
> Recall that in our human evaluation, 67% of the time sketches from our Creative Birds datasets were chosen to be more creative than the QuickDraw dataset. This demonstrates a qualitative difference between our dataset and QuickDraw.
>
> Finally, we did some additional quantitative analysis. As a proxy for complexity, we report the average stroke length (normalized by the image size) and number of strokes across the bird sketches in each dataset. We see that our datasets have the longest and most number of strokes. The difference is especially stark w.r.t. QuickDraw (twice as long and three times as many strokes), one of the most commonly used sketch datasets. We have added this to Appendix D.
>
> ```
>                         norm. stroke length  num. of stroke
> ------------------------------------------------------------
>  Sketchy                    5.54 ± 2.02      15.57 ± 10.54
>  Tu-Berlin                  5.46 ± 1.75      14.94 ± 10.18
>  QuickDraw                  4.00 ± 2.97       6.85 ± 4.19
>  Creative Birds (Ours)      7.01 ± 2.89      20.74 ± 13.37
>  Creative Creatures (Ours)  8.40 ± 3.47      22.65 ± 14.73
> ```
>
> ***“I think a user-in-the-loop interface is a reasonable approach, but though the model seems compatible with such a system, the authors do not actually implement it”***
>
> We could not agree more that our method is a natural fit for a human-in-the-loop application! We did implement this. You can take a look at a screen capture here https://streamable.com/zygr8h. This human-in-the-loop demo will be made publicly available.
>
> ***“it would be useful to have information about the order of individual strokes, which the dataset does not appear to include.”***
>
> Our datasets do include order information of the individual strokes. Each sketch in the dataset is stored as a vector image. We also model this temporal information at the part-level (part selector in Figure 3 in the paper).
>
> ***“I'm not sure about the argument that raster images are a better format for a generative model than vector. It seems like the long-range relationships and connectivity as well as the sparsity inherent in a vector representation are pretty important in a sketch”***
>
> We agree that vector representations are worth exploring further. Their inherently sparse nature and arbitrary resolution are strong points in their favor. That said, we do think that the long range interaction between strokes often boils down to spatial relationships between different parts of the sketch. Those may be more naturally modeled in raster image representations (as opposed to long range interactions in a temporal sequence). Note that our generations can be readily converted to vector representations to remove some of the artifacts of "poorly defined line segments" as shown in Appendix L.
>
> Of course, much of this is speculation and it is certainly worth continuing to explore other representations for this task. A hybrid approach that models shape and positions of parts in the spatial domain and then generates the sketch in vector representations seems promising for future work.
>
> ***“How were the 100 sketches from Creative Birds and QuickDraw chosen for the user study?”***
>
> They were picked randomly.
>
> ***“The paper claims the importance of certain architecture choices, e..g, that part channels help the model better predict the part locations. Some ablation study to verify this would be appropriate.”***
>
> Agreed. Note that without some of these architectural modifications such as the sparsity loss, the model completely breaks down (e.g., does not train, generates blank sketches).
>
> Regarding part channels specifically: We analyzed their importance for head generation. Given a generated eye, the head is expected to surround the eye. Out of 512 random generations, this was the case 57.4% of the time without using part channels, and 88.7% with part channels. We have added this to Appendix K.

---

> > ### Comment · AnonReviewer1 · 2020-11-24
> > **Response**
> >
> > Thank you for your response. I still am skeptical that the proposed dataset is a fundamental improvement over Quick Draw---ultimately, both datasets contain compact sketches from a single category containing relatively few strokes. But given that your updates and clarifications have addressed many of my questions, I am raising my score.

---

### Author Response · Authors · 2020-11-18
**General Response**

We would like to thank the reviewers for their thoughtful feedback. We are glad to see that reviewers generally appreciated the contributions of our paper -- the interestingness of the creative sketch generation task (R3, R4), the novelty of our dataset collection protocol (R1) and the value of our datasets to future research (R1, R2, R3, R4), the design of our model (R4) and its usefulness in this application (R1), solid experimental results (R2, R4), useful evaluation metrics (R2) and writing clarity (R1, R3, R4). Below, we address each reviewer’s comments.

---

### Decision · Program_Chairs · 2021-01-07
**Final Decision**

**Decision:**

Accept (Poster)

**Comment:**

While much of generative modeling is tasked with the goal of generating content within the data distribution, the motivation of this work is to examine whether ML techniques can generate creative content. This work has 2 core contributions:

1) Two new datasets of creative sketches: birds and creatures, that have part annotations (size ~ 10K samples for each set). The way the datasets are structured with the body part annotations will facilitate the creativity aspect of the approach later described in the paper.

2) This paper propose a GAN model that is part-based, which they call DoodlerGAN. It is inspired partly by the human's creative process of sequential drawing. Here, the trained model determines the appropriate order of parts to generate, which makes the model well suited for human-in-the-loop interactive interfaces in creative applications where it can make suggestions based on user drawn partial sketches.

They show that the proposed model, trained on the part-annotated datasets, are able to generate unseen compositions of birds and creatures with novel body part configurations for creative sketches. They conduct human evaluation and also quantitative metrics to show the superiority of their approach (for human preference, and also FID score).

Many reviewers, including R1 and myself observe that the datasets provided, along with the parts-based labeling and modeling approach are a clear advantage over existing datasets and methodology. With ever growing importance of generative models used in real world applications, including the creative industry, I believe this paper provides a much needed fresh take on creative ways of using our generative models besides making them larger, or achieving better log-likelihood scores. Many reviewers, including R3, would think that this work is indeed a "Delightful, well written paper! I have concerns about its fit here." I strongly believe such works in fact definitely *do* belong at ICLR, and I think this work has the potential to get researchers in the generative modeling community to rethink what they are really optimizing for.

I believe this paper will be a great addition to ICLR2021, and I look forward to see their presentation to the community to spark more creativity in our research endeavors. For this reason, I'm strongly recommending an acceptance (Poster).